# Improving the safety of human pluripotent stem cell therapies using genome-edited orthogonal safeguards

Renata M. Martin [1,2,6], Jonas L. Fowler[1,3,6], M. Kyle Cromer[1,2], Benjamin J. Lesch[1,2], Ezequiel Ponce[1,2], Nobuko Uchida[1,2,4], Toshinobu Nishimura [1,5], Matthew H. Porteus [1,2,7✉] & Kyle M. Loh [1,3,7✉]

Despite their rapidly-expanding therapeutic potential, human pluripotent stem cell (hPSC)-derived cell therapies continue to have serious safety risks. Transplantation of hPSC-derived cell populations into preclinical models has generated teratomas (tumors arising from undifferentiated hPSCs), unwanted tissues, and other types of adverse events. Mitigating these risks is important to increase the safety of such therapies. Here we use genome editing to engineer a general platform to improve the safety of future hPSC-derived cell transplantation therapies. Specifically, we develop hPSC lines bearing two drug-inducible safeguards, which have distinct functionalities and address separate safety concerns. In vitro administration of one small molecule depletes undifferentiated hPSCs $>10^6$-fold, thus preventing teratoma formation in vivo. Administration of a second small molecule kills all hPSC-derived cell-types, thus providing an option to eliminate the entire hPSC-derived cell product in vivo if adverse events arise. These orthogonal safety switches address major safety concerns with pluripotent cell-derived therapies.

[1] Institute for Stem Cell Biology & Regenerative Medicine, Stanford University School of Medicine, Stanford, CA 94305, USA. [2] Department of Pediatrics, Stanford University School of Medicine,  Stanford, CA 94305, USA. [3] Department of Developmental Biology, Stanford-UC Berkeley Siebel Stem Cell Institute, Stanford Ludwig Center for Cancer Stem Cell Research and Medicine, Stanford University School of Medicine, Stanford, CA 94305, USA. [4] ReGen Med Division, BOCO Silicon Valley, Palo Alto, CA 94303, USA. [5] Department of Genetics, Stanford University School of Medicine, Stanford, CA 94305, USA. [6] These authors contributed equally: Renata M. Martin, Jonas L. Fowler. [7] These authors jointly supervised this work: Matthew H. Porteus, Kyle M. Loh. ✉email: mporteus@stanford.edu; kyleloh@stanford.edu

ncreasing numbers of human pluripotent stem cell (hPSC)-derived cell therapies have been transplanted into patients, with over 30 ongoing or completed clinical trials for multiple indications, including spinal cord injury, macular degeneration, and type 1 diabetes[1]. The breadth of these clinical trials highlights the promise of hPSC-derived cell therapies. However, hPSC-based therapies present unique safety risks compared to adult-derived cell therapies[2,3]. To realize the potential of hPSC-derived therapies, strategies to mitigate these unique risks need to be further developed. These risks fall into two main categories (Fig. 1a).

First, hPSC differentiation often yields a heterogeneous cell population[4,5], and even a small number of residual undifferentiated hPSCs (10,000 or even fewer) can form a teratoma in vivo[6,7]. If billions of hPSC-derived cells are to be transplanted into a patient, even 0.001% remaining hPSCs might be therapeutically unacceptable; thus a 5-log depletion of undifferentiated hPSCs will be critical[5]. Indeed, transplantation of certain hPSC-derived liver[8] and pancreatic[9–11] populations yielded teratomas in animal models[3], which would be concerning if they similarly arose in human patients.

Second, differentiated cell-types of the wrong lineage can, upon transplantation, generate tumors or unwanted tissues altogether. For example, transplantation of PSC-derived neural populations into animal models generated tumors[12–14] or cysts[15] in some cases. Indeed hPSCs have also been reported to acquire certain genetic abnormalities in culture (e.g., *TP53* mutations or *BCL2L1* amplifications)[16–18], some of which induce their differentiated progeny to form tumors in vivo[13]. These safety issues may be further exacerbated as hPSCs are engineered to be hypoimmunogenic in order to minimize their rejection by patients' immune systems[19,20]. Notably, if hPSC-derived hypoimmunogenic cells become malignantly transformed or virally infected, they may not be adequately controlled by the recipient's immune system. In such cases, an inducible system to eliminate all transplanted hPSC-derived cells would be a valuable tool to reduce these risks.

To mitigate both of these safety risks for hPSC-based cell therapies, here we develop orthogonal systems to selectively kill undifferentiated hPSCs or to efficiently eliminate the entire cell product if necessary (Fig. 1b–d). All three of these genetically encoded safety systems ($NANOG^{iCaspase9}$, $ACTB^{OiCaspase9}$, and $ACTB^{TK}$) are specifically inserted into endogenous genes, and enable us to ablate hPSC-derived cell populations upon small molecule administration both in vitro and in vivo.

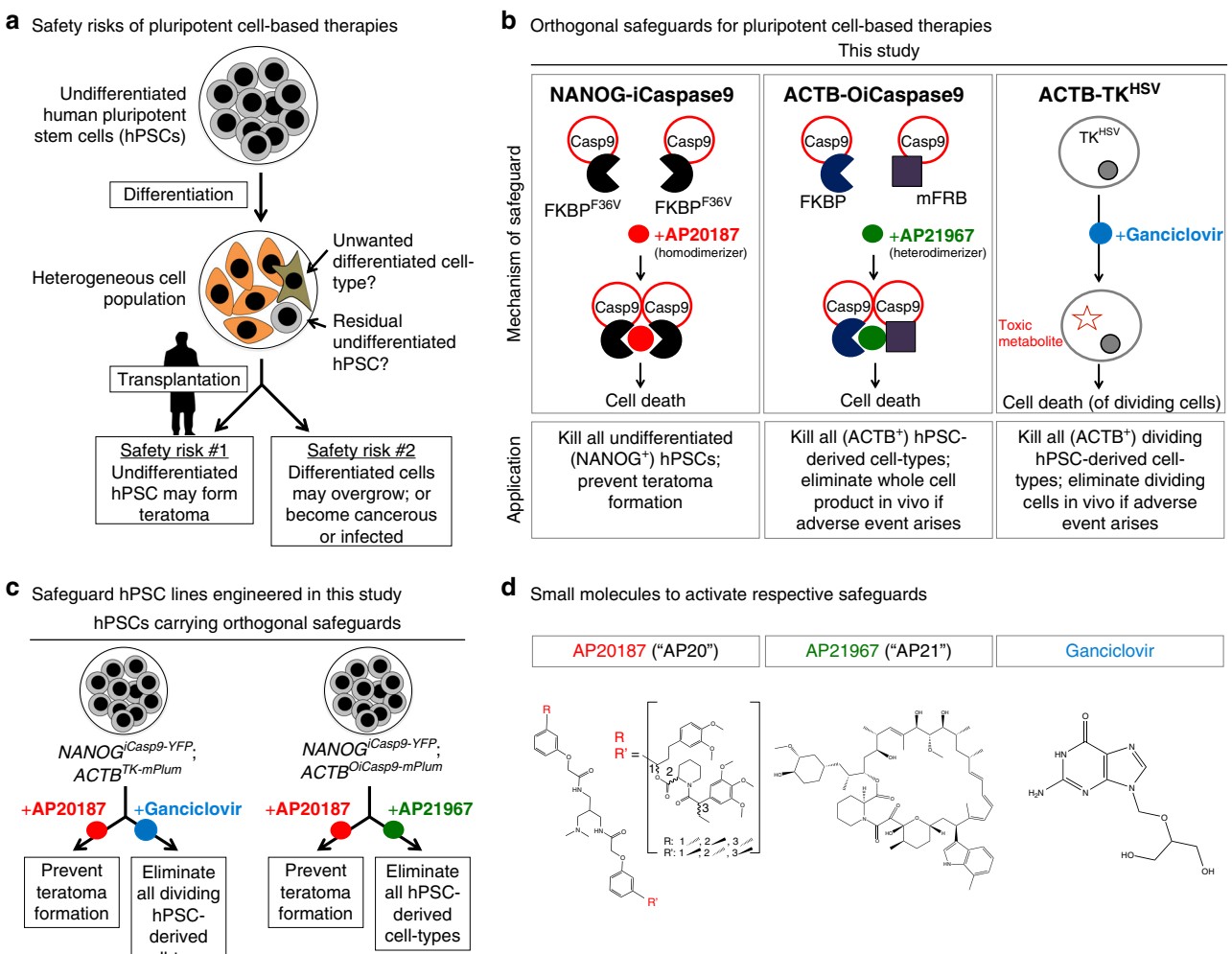

**Fig. 1 Genetically engineered safeguards for human pluripotent stem cell-based therapies. a** Safety risks of human pluripotent stem cell (hPSC)-based cell therapies. **b** Summary of the safeguards described in this study. iCaspase9 inducible Caspase9, OiCaspase9 orthogonal inducible Caspase9, $TK^{HSV}$ herpes simplex virus-derived thymidine kinase, $FKBP^{F36V}$ FKBP12 with F36V point mutation, mFRB mutant FRB domain. **c** Applications of the safeguards described in this study. **d** Small molecules used to activate respective safeguards.

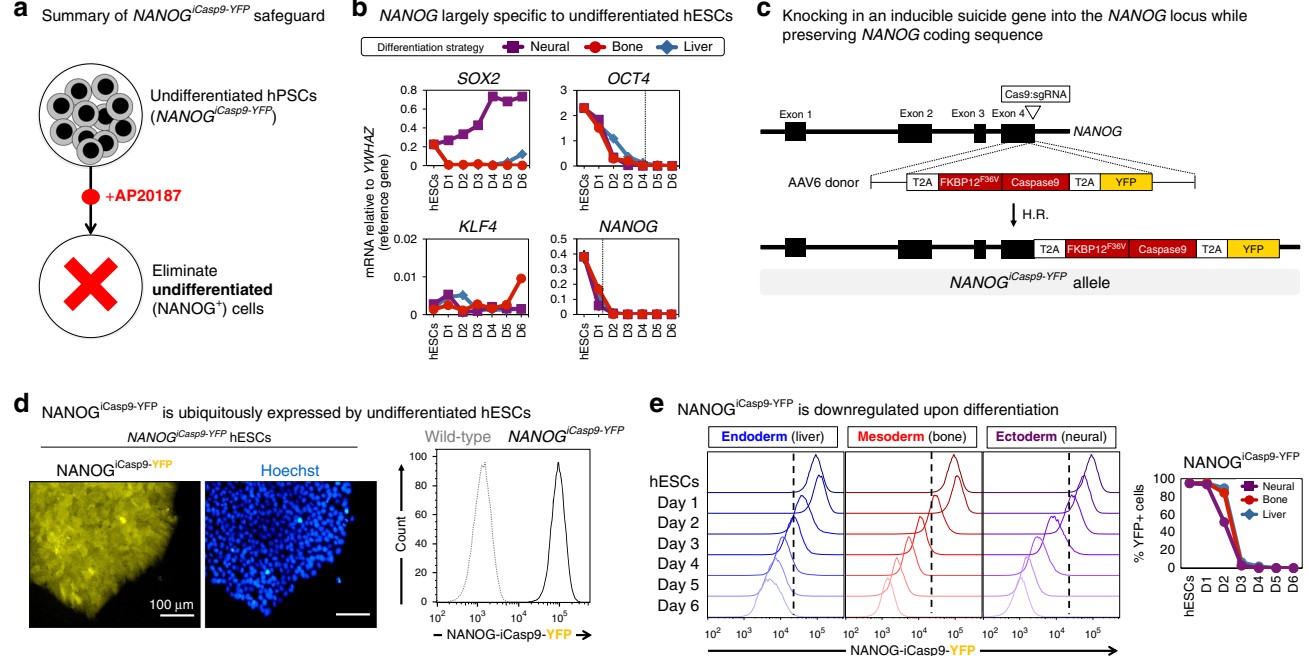

**Fig. 2 Rationale and design of the *NANOG^iCasp9-YFP* safety switch. a** Intended application of the *NANOG^iCasp9-YFP* safeguard. **b** Quantitative PCR (qPCR) of pluripotency transcription factor expression during differentiation into endodermal[29,30], mesodermal[31], and ectodermal[32] lineages (differentiation was conducted as described in the "Methods"). Dotted line indicates when gene expression declined below 10% of *YWHAZ* in all three differentiation systems. Expression of lineage markers is depicted normalized to the reference gene *YWHAZ* (i.e., *YWHAZ* = 1.0). Error bars = standard error. **c** Cas9 RNP/AAV6-based strategy for *NANOG^iCasp9-YFP* targeting[38]. **d** YFP expression levels in *NANOG^iCasp9-YFP* hESCs as shown by epifluorescence (*left*) and flow cytometry (*right*), relative to wild-type hESCs. Scale bar = 100 μm. **e** Flow cytometric analysis of YFP during differentiation of *NANOG^iCasp9-YFP*; *ACTB^TK-mPlum* hESCs into endodermal, mesodermal, or ectodermal lineages. Dotted line delineates negative vs. positive cells set based on YFP levels in undifferentiated *NANOG^iCasp9-YFP*; *ACTB^TK-mPlum* hESCs. Error bars = standard deviation. Source data are available in the Source Data file.

## Results

**Selectively killing undifferentiated hPSCs by *NANOG^iCaspase9*.**
We addressed the safety concern that trace numbers of undifferentiated hPSCs can form teratomas in vivo[6,8–11] (Figs. 1 and 2a). Others have described surface markers that identify undifferentiated hPSCs (e.g., SSEA-3, SSEA-4, TRA-1-60, TRA-1-81, and PODXL1)[21,22] as well as genetic kill-switches (based on expression of the *CDK1*[23], *TERT*[24], *SCD1*[25], *SURVIVIN/BIRC5*[26,27], and *SOX2*[28] genes) to kill such cells. However, the efficacy of all such systems depends on whether expression of these marker genes is specific to pluripotent cells. We found that all of these previously reported markers were expressed by undifferentiated hPSCs as well as by cells that had been differentiated into endoderm (liver progenitors[29,30]), mesoderm (bone progenitors[31]), and ectoderm (forebrain progenitors[32]) (Supplementary Fig. 1a–c, Fig. 2b). Hence, previous marker-based strategies to deplete pluripotent cells are not specific and would also deplete the therapeutic product consisting of differentiated cells. Indeed we found that the small-molecule SURVIVIN inhibitor YM155 (which was previously reported to kill undifferentiated hPSCs[26,27]) was inhibitory to the growth of both undifferentiated and differentiated hPSCs (Supplementary Fig. 1d), consistent with broad expression of *SURVIVIN* across undifferentiated and differentiated hPSCs (Supplementary Fig. 1a). This emphasizes the importance of selectively depleting undifferentiated hPSCs to create a safe differentiated cell product that could then be safely transplanted with a significantly decreased risk of teratoma formation.

We assayed the expression of multiple pluripotency transcription factors[33] and found that *NANOG* was the most specific to the pluripotent state (Fig. 2b). *NANOG* is crucial for pluripotency in human and mouse and its expression is largely restricted to pluripotent cells in vivo[34–37]. Indeed we found that *NANOG* was expressed by undifferentiated hPSCs but was sharply downregulated within 24 h of ectoderm differentiation and within 48 h of endoderm or mesoderm differentiation in vitro (Fig. 2b).

We therefore developed a specific and simple system to track whether cells were in a pluripotent state (*NANOG^+*) and to link this to controllable elimination of such cells via apoptosis. We exploited Cas9 RNP (ribonucleoprotein)/AAV6-based genome editing[38] to knock-in an inducible Caspase9 (*iCaspase9*) cassette[39] and a fluorescent reporter (*YFP*) immediately downstream of the *NANOG* coding sequence (Fig. 2c, Supplementary Fig. 2a), thus creating a *NANOG^iCasp9-YFP* knock-in allele while leaving the *NANOG* coding sequence intact, as *NANOG* is critical for pluripotency[34,35,37]. The *NANOG*, *iCaspase9*, and *YFP* genes are all transcribed together from the *NANOG^iCasp9-YFP* allele but are separated by T2A self-cleaving peptides[40] such that after translation, they are expressed as three separate proteins. *iCaspase9* encodes a Caspase9-FKBP^F36V fusion protein that, after dimerization with the small molecule AP20187 (hereafter called "AP20"), induces cell-intrinsic, rapid, and irreversible apoptosis (Fig. 1b)[39]. hPSCs should not be able to silence this knock-in *NANOG^iCasp9-YFP* system, because if they downregulated endogenous *NANOG* expression, they would no longer be pluripotent[37]. Importantly, we inserted the *NANOG^iCasp9-YFP* allele into both *NANOG* loci to prevent the emergence of escape cells (e.g., if a pluripotent cell stochastically used only one allele of *NANOG* to support its growth and pluripotent state[41]).

Genomic sequencing confirmed successful biallelic targeting of the *NANOG* locus, without off-target integration into the *NANOGP8* pseudogene (Supplementary Fig. 2a). *NANOG^iCasp9-YFP* hPSCs maintained normal pluripotency marker expression (Supplementary Fig. 2b and c), karyotype (Supplementary Fig. 2d) and the ability to differentiate into endoderm, mesoderm, and ectoderm cells

(Supplementary Fig. 1c). NANOG protein and mRNA were expressed at normal levels in *NANOG$^{iCasp9-YFP}$* hPSCs (Supplementary Fig. 2b and c), showing that insertion of the *iCasp9-YFP* cassette downstream of the *NANOG* gene did not noticeably affect *NANOG* expression. The *NANOG$^{iCasp9-YFP}$* allele faithfully paralleled endogenous *NANOG* expression: YFP and *iCaspase9* mRNA were uniformly expressed by undifferentiated *NANOG$^{iCasp9-YFP}$* hPSCs, but both were extinguished upon endoderm, mesoderm, or ectoderm differentiation (Fig. 2d, e, Supplementary Fig. 2e). The *NANOG$^{iCasp9-YFP}$* knock-in allele remained active in undifferentiated hPSCs even after long-term culture, demonstrating that it was not silenced (Supplementary Fig. 2f). After successfully engineering the cells, we tested whether the *NANOG$^{iCasp9-YFP}$* system could specifically ablate undifferentiated hPSCs without eliminating differentiated cells (the potential therapeutic cell product).

AP20 treatment activated iCaspase9 in undifferentiated *NANOG$^{iCasp9-YFP}$* hPSCs and eliminated them, while sparing their differentiated progeny (Fig. 3). This system was effective (depleting undifferentiated hPSCs $1.75 \times 10^6$-fold), sensitive (activated by 1 nM AP20), and specific (sparing > 95% of differentiated bone, liver, or forebrain progenitors). Twenty-four hours treatment with 1 nM of AP20 led to a $1.75 \times 10^6$-fold depletion of undifferentiated hPSCs (as assayed across seven independent experiments; Fig. 3a). This *NANOG$^{iCasp9-YFP}$* system thus enables greater than the 5-log reduction of hPSCs anticipated to be needed to ensure safety of a cell product with a billion differentiated cells. It also demonstrates quantitative killing of hPSCs exceeding prior reported systems, which generally deplete undifferentiated hPSCs by 1-log or less[22,24–27]. AP20 was remarkably potent ($IC_{50} = 0.065$ nM (Supplementary Fig. 3a, b)) and rapid (even 12 h of treatment sufficed to eliminate hESCs (Supplementary Fig. 3c)). 1 nM AP20 was the optimal dose to activate the *NANOG$^{iCasp9-YFP}$* system (Fig. 3a), as higher AP20 concentrations downregulated *NANOG* (Supplementary Fig. 3b). AP20 is structurally related to FK506

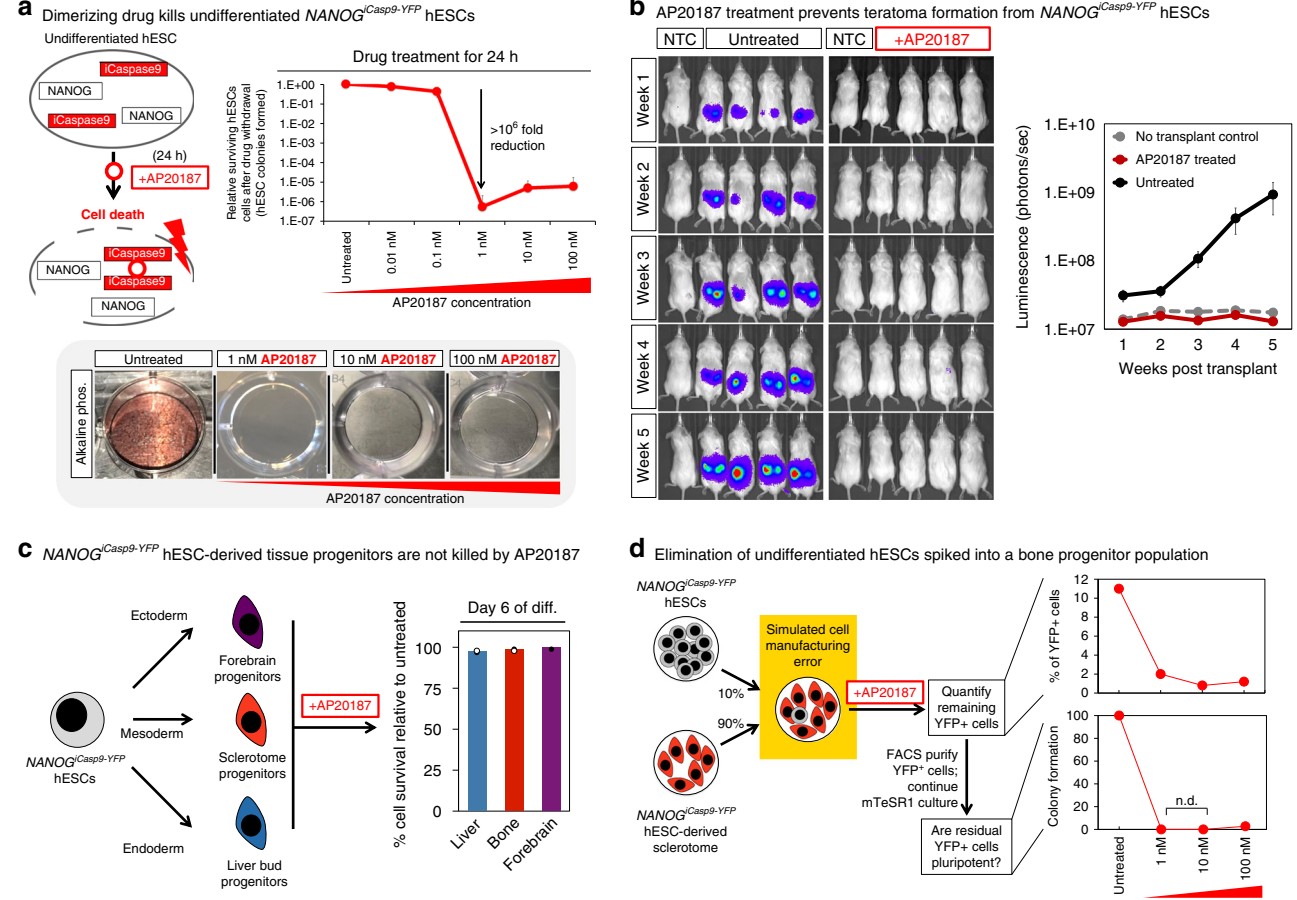

**Fig. 3 Implementation of the *NANOG$^{iCasp9-YFP}$* safety switch. a** Schema depicting how drug AP20187 induces dimerization of Caspase9 in undifferentiated *NANOG$^{iCasp9-YFP}$* hESCs, subsequently triggering cell death (top left). *NANOG$^{iCasp9-YFP}$* hESCs were treated for 24 h with increasing concentrations of AP20187. AP20187 was withdrawn, and cultures were further grown in mTeSR1 to allow any surviving hESCs to grow; any surviving colonies were counted 1 week later (top right). Alkaline phosphatase staining of whole wells of a 12-well plate after 24-h treatment with AP20187 (bottom). Error bars = standard deviation. **b** $5 \times 10^5$ *NANOG$^{iCasp9-YFP}$* hESCs engineered to express AkaLuciferase[43] were treated with control media or 1 nM AP20187 for 24 h, and then subcutaneously transplanted into the right and left dorsal flanks of NOD-SCID *Il2rg$^{-/-}$* mice ($5 \times 10^5$ cells per flank). Bioluminescent imaging of mice occurred weekly for 5 weeks with no transplant control (NTC) animals included for relative luminescence normalization. Total flux (photons/s) was measured for each animal and averaged. Error bars = standard deviation. **c** *NANOG$^{iCasp9-YFP}$* hESCs were differentiated for 6 days into derivatives of the ectoderm (forebrain), mesoderm (sclerotome), and endoderm (liver bud); for the last 24 h, they were treated with 1 nM AP20187. The percentage of surviving cells was calculated relative to untreated controls. Error bars = standard deviation. **d** *NANOG$^{iCasp9-YFP}$* hESCs were mixed 1:9 with *NANOG$^{iCasp9-YFP}$* hESC-derived day-5 sclerotome cells and were cultured for 24 h in sclerotome media (supplemented with 100 ng/mL FGF2 to help hESCs maintain pluripotency; described in "Methods"), in the presence or absence of AP20187. Flow cytometric analysis was done to determine the percentage of YFP+ hESCs left in the mixed population (top right). Surviving YFP+ hESCs were FACS sorted and cultured in mTeSR1 for 1 week to determine whether they were still capable of forming colonies (*bottom right*). Source data are available in the Source Data file.

(which is also a BMP agonist)[42], and BMP activation is known to downregulate *NANOG* in hPSCs[37].

Given that very small numbers of hPSCs (~10,000) are sufficient to form teratomas in vivo[6], we tested the lower bounds of this system to test whether any rare hPSCs survived drug treatment and whether they could form teratomas. We pre-treated $5 \times 10^5$ hPSCs with control media or 1 nM AP20 for 24 h prior to subcutaneous transplantation into the left and right dorsal flanks of NOD-SCID $Il2rg^{-/-}$ (NSG) mice to form teratomas (Fig. 3b). Past studies primarily assessed teratoma formation through visual inspection of subcutaneous nodules[23–25,27]. However, here we used bioluminescent imaging to determine if micro-teratomas might form, because *AkaLuciferase*-based bioluminescent imaging is more quantitative and sensitive; it can detect a single cell in vivo[43]. Bioluminescent imaging revealed that 0/19 of mice transplanted with AP20-treated hPSCs formed teratomas, whereas 19/19 of mice transplanted with control-treated hPSCs formed teratomas ($N = 3$ independent experiments; Fig. 3b). Taken together, the ability to prevent the formation of even microscopic teratomas is an important step towards developing safer pluripotent cell-derived therapies.

Importantly, 24-h treatment with AP20 specifically eliminated undifferentiated hPSCs while sparing differentiated hPSC-derived tissue progenitors: >95% of $NANOG^{iCasp9-YFP}$ hPSC-derived day-6 liver progenitors[29,30], bone progenitors[31], and forebrain progenitors[32] all remained viable (Fig. 3c, Supplementary Fig. 3d), and expression of differentiation markers was not substantially affected (Supplementary Fig. 3e). This is consistent with the absence of NANOG expression in each of these hPSC-derived tissue progenitor populations (Fig. 2b, e). The $NANOG^{iCasp9-YFP}$ system also specifically eliminated undifferentiated hPSCs within heterogeneous cell populations. To simulate a cell-manufacturing failure, we generated day-5 hPSC-derived bone (sclerotome) progenitors and deliberately introduced 10% undifferentiated hPSCs (Fig. 3d). Treatment with AP20 for the last 24 h of differentiation led to a >10-fold decrease in NANOG-YFP$^+$ cells (monitored by virtue of the YFP encoded by the $NANOG^{iCasp9-YFP}$ allele) (Fig. 3d). The surviving NANOG-YFP$^+$ cells were compromised and were no longer pluripotent, as upon FACS purification and continued culture in hPSC media, they did not form colonies within the limit of detection of our assay (Fig. 3d). Similar results were observed when mixing hPSCs and sclerotome cells at different ratios (Supplementary Fig. 3f). In conclusion, AP20 treatment of $NANOG^{iCasp9-YFP}$ hPSCs provides an effective, sensitive, rapid, and selective means to eliminate undifferentiated hPSCs without eliminating differentiated progeny.

**Eliminating hPSC-derived cell populations in vivo by $ACTB^{TK}$.** While the $NANOG^{iCasp9-YFP}$ system reduces teratoma risk, this is not the only concern for hPSC-derived cell therapies, as differentiated PSC-derived cell-types can uncontrollably proliferate in vivo, as observed for neural tumors[12–14]. The $NANOG^{iCasp9-YFP}$ system would not be an effective safeguard for this type of adverse event. We thus developed an orthogonal drug-inducible safeguard to curb the growth of, or eliminate, all transplanted cells in vivo if overgrowing, unwanted, or damaging tissues/cells are detected post-transplantation (Fig. 4a). This system could also be used to eliminate transplanted hPSC-derived cells once their therapeutic effect was achieved, thus allowing a living drug to have a controllable endpoint.

To this end, in the $NANOG^{iCasp9-YFP}$ hPSC line, we knocked-in a second drug-inducible kill-switch (herpes simplex virus-derived thymidine kinase (TK))[44] and a fluorescent reporter (mPlum) into a constitutively expressed gene (ACTB [BETA-ACTIN]), thus

creating a $ACTB^{TK-mPlum}$ knock-in allele (Fig. 4b, Supplementary Fig. 4a). The *ACTB*, *TK*, and *mPlum* genes are all transcribed together from this allele but are separated by T2A self-cleaving peptides[40] such that after translation, they are expressed as three separate proteins. TK phosphorylates ganciclovir to a nucleotide analog that competes with ddGTP which, after incorporation into DNA during replication, results in chain termination, consequently killing dividing cells[44]. Ganciclovir treatment should therefore eliminate all hPSC-derived dividing cell-types, irrespective of their lineage or differentiation status (since the *ACTB* gene is ubiquitously expressed and is essential for proliferation; Supplementary Fig. 4b). The proliferation, pluripotency marker expression, karyotype, and differentiation potential of $ACTB^{TK-mPlum}$;$NANOG^{iCasp9-YFP}$ hPSCs was not perturbed (Fig. 4c, Supplementary Figs. 1c and 2b–d).

The $ACTB^{TK-mPlum}$ cassette was highly expressed in undifferentiated hPSCs as well as hPSC-derived endoderm, mesoderm, and ectoderm tissue progenitors (Fig. 4c, d), paralleling *ACTB* mRNA expression (Supplementary Fig. 4b). Because TK is expressed under the control of the endogenous *ACTB* locus, our system should evade silencing, unlike previous transgenes driven by exogenous viral promoters[45,46]. Indeed, given that *ACTB* is generally an essential gene[47], if the allele was silenced, the cell would not proliferate and would die.

We found that ganciclovir treatment completely eliminated $ACTB^{TK-mPlum}$;$NANOG^{iCasp9-YFP}$ hPSCs (Fig. 5a) and substantially eliminated within 24 h of drug exposure their derivative liver, bone, and forebrain progenitors (Fig. 5b). The differing efficacy of $ACTB^{TK-mPlum}$ in undifferentiated and differentiated hPSCs may relate to the differing proliferative rates of these lineages. To eliminate all cells independent of their division rate, we developed a separate kill-switch ($ACTB^{OiCaspase9}$; described in a subsequent section [Figs. 1 and 6]).

We tested the $ACTB^{TK-mPlum}$ safeguard in an in vivo model in which growth of hPSC-derived tissues had to be suppressed or even potentially eliminated (Fig. 5c). In this model system, we subcutaneously transplanted undifferentiated $10^6$ $ACTB^{TK-mPlum}$;$NANOG^{iCasp9-YFP}$ hPSCs into the left and right dorsal flanks of NSG mice, which formed teratomas in 3 weeks (Fig. 5c, Supplementary Fig. 4c). Starting at 3 weeks post-transplantation, we treated transplanted mice with ganciclovir every day. This ablated any detectable teratomas as measured by bioluminescent imaging: by week 7 post-transplantation (i.e., 4 weeks after initiating ganciclovir administration), 10/10 of control mice harbored detectable teratomas, whereas 0/10 ganciclovir-treated mice had detectable teratomas (Fig. 5c, Supplementary Fig. 4c). The complete elimination of teratomas by our $ACTB^{TK-mPlum}$ system contrasts with results obtained with a past $CDK1^{TK-mCherry}$ system, which incompletely suppressed teratoma growth[23].

Finally, we confirmed that the $NANOG^{iCasp9-YFP}$ system (which is activated by AP20) was still operational in the dual-safeguard $NANOG^{iCasp9-YFP}$;$ACTB^{TK-mPlum}$ hPSC line. To demonstrate this, we pre-treated the dual $NANOG^{iCasp9-YFP}$;$ACTB^{TK-mPlum}$ hPSCs with 1 nM AP20 before transplantation (to activate the $NANOG^{iCasp9-YFP}$ system), which prevented them from forming teratomas in vivo (Fig. 5c, Supplementary Fig. 4c).

In sum, this series of experiments demonstrate the power of the $ACTB^{TK-mPlum}$ safety switch: treatment with ganciclovir can be used to eliminate undesired hPSC-derived cell populations in vivo. However, the TK system took a prolonged amount of time (~1 month) to eliminate teratomas (Fig. 5c, Supplementary Fig. 4c) and it principally kills dividing cells (although bystander cells may also be indirectly eliminated)[44]. A different tool to rapidly kill the entire hPSC-derived cell product (not just dividing cells) would thus be a further advance, and is described below.

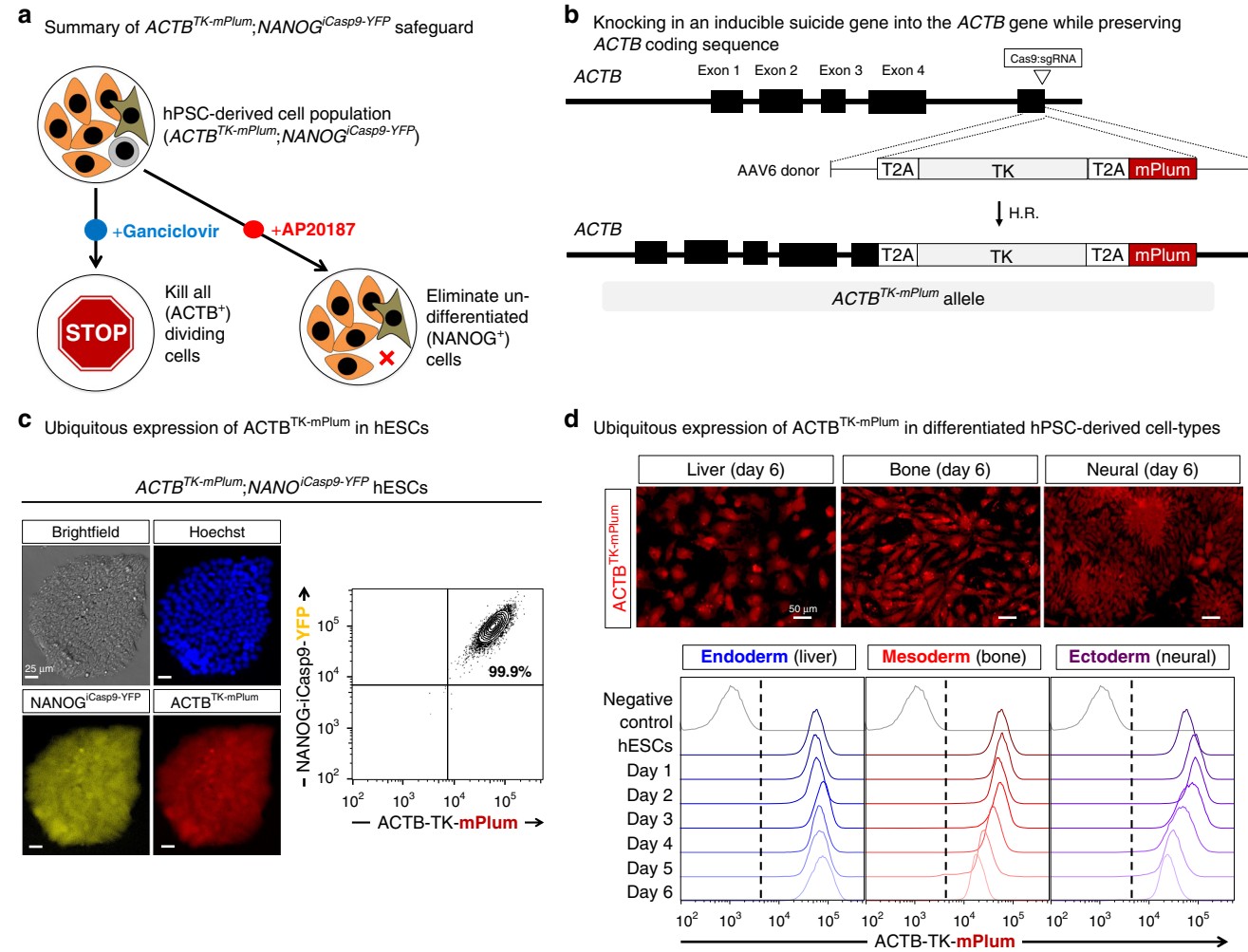

**Fig. 4 Rationale and design of the *ACTB^TK-mPlum* safety switch. a** Intended application of the dual *NANOG^iCasp9-YFP*;*ACTB^TK-mPlum* safeguard. **b** Cas9 RNP/ AAV6-based knock-in strategy for *ACTB^TK-mPlum* targeting[38], which was performed in the *NANOG^iCasp9-YFP* hESC line. **c** mPlum was highly expressed in undifferentiated *NANOG^iCasp9-YFP*;*ACTB^TK-mPlum* hESCs, as shown by epifluorescence (*left*) and flow cytometry (*right*). Scale bar = 25 μm. **d** *NANOG^iCasp9-YFP*; *ACTB^TK-mPlum* hESCs were differentiated into day 6 liver, sclerotome, and neural progenitors with mPlum levels remaining high throughout each type of differentiation as shown by epifluorescence (*top*, for day 6 progenitors) and flow cytometry (*bottom*, each 24 h of differentiation). Dotted line delineates negative versus positive cells, with the gate set on negative control (wild-type) hESCs. Scale bar = 50 μm.

**Engineering an orthogonal iCaspase9**. To rapidly kill the entire hPSC-derived cell product, we created an orthogonal iCaspase9-based killing system that would be compatible with our *NANO-G^iCasp9-YFP* system (Fig. 1), which specifically eliminates undifferentiated hPSCs. While iCaspase9 dimerization is induced by AP20 (resulting in apoptosis)[39], we engineered a variant of iCaspase9 that can be activated by a second orthogonal small molecule that is not AP20 (Figs. 1b and 6a).

This orthogonal iCaspase9 (henceforth, OiCaspase9) comprises Caspase9 fused to both a mutant FRB domain and a FKBP domain; these two domains are dimerized by a different small molecule (AP21967, hereafter called "AP21")[48] (Fig. 6a). To implement and test this OiCaspase9 system, we knocked it into the *ACTB* gene in *NANOG^iCasp9-YFP* hPSCs (Fig. 6b; Supplementary Fig. 5a), thus generating *ACTB^OiCasp9-mPlum*;*NANOG^iCasp9-YFP* hPSCs that were karyotypically normal (Supplementary Fig. 5b). In the *ACTB^OiCasp9-mPlum* knock-in allele, the *ACTB*, *OiCasp9*, and *mPlum* genes are separated by T2A self-cleaving peptides[40], such that they are transcribed together but are expressed as three separate proteins (Fig. 6b).

**Directly killing all hPSC-derived cells by *ACTB^OiCaspase9***. In this dual-safeguard (*ACTB^OiCasp9-mPlum*;*NANOG^iCasp9-YFP*) hPSC line, we could kill all hPSC-derived cell-types (whether undifferentiated or differentiated liver, bone, and neural progenitors) through treatment with AP21, which activated the *ACTB^OiCasp9-mPlum* kill-switch (Fig. 6c–e). Alternatively, we could selectively kill undifferentiated hPSCs through treatment with AP20, which activated the *NANOG^iCasp9-YFP* kill-switch (Fig. 6e).

Importantly, iCaspase9 and OiCaspase9 did not cross-react: AP21 selectively activated OiCaspase9 (but not iCaspase9; Fig. 6d, Supplementary Fig. 5c and d). Reciprocally, AP20 specifically activated iCaspase9 (but not OiCaspase9; Fig. 6e). Therefore, iCaspase9 and OiCaspase9 constitute orthogonal kill-switches, providing a toolkit to inducibly kill distinct cell subsets in response to different contingencies: for instance, (1) selectively eliminating undifferentiating hPSCs to reduce teratoma risk using AP20 (which activates *NANOG^iCasp9-YFP*) or (2) killing all hPSC-derived cell-types if an adverse event arises using AP21 (which activates *ACTB^OiCasp9-mPlum*) (Fig. 1a).

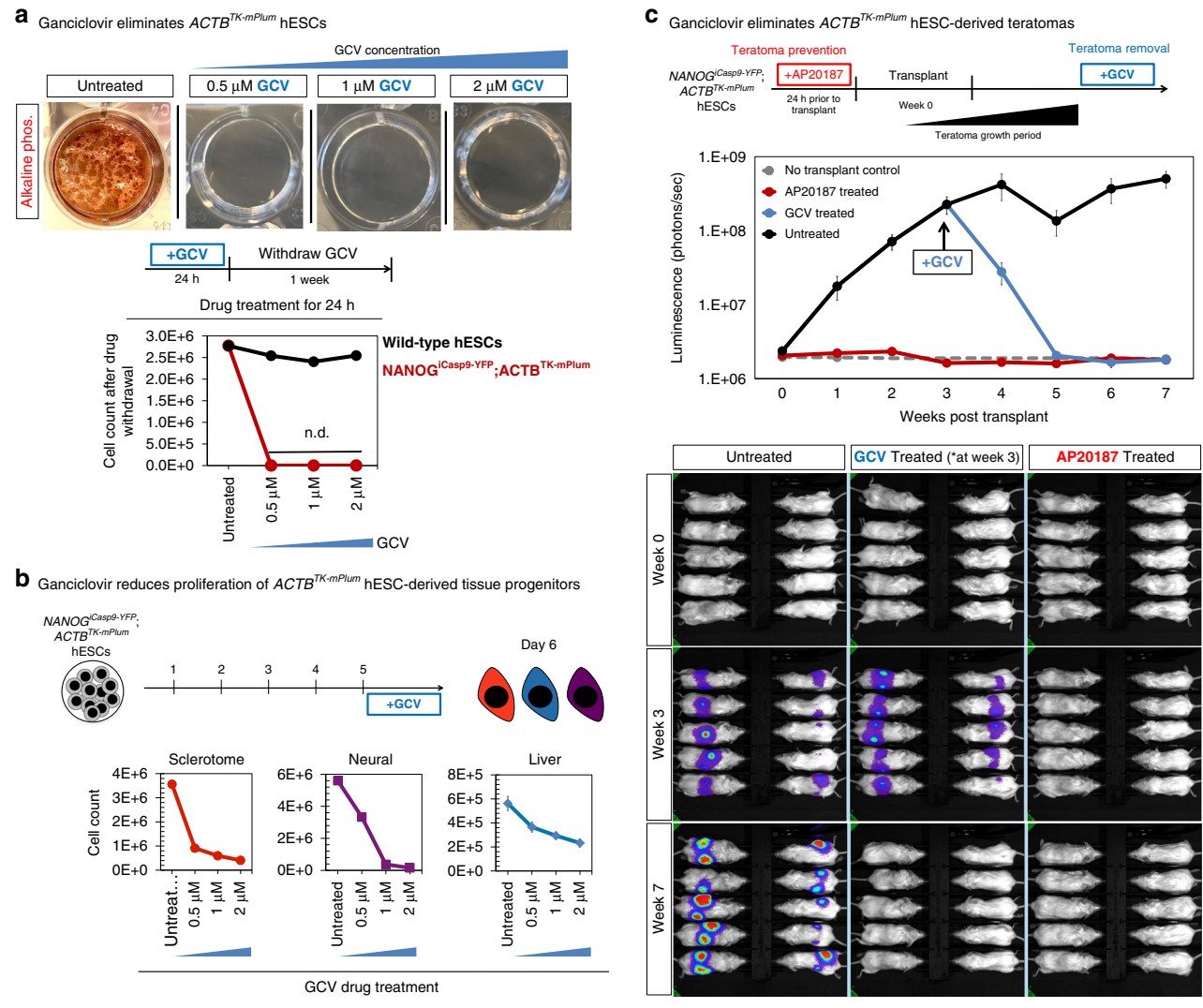

**Fig. 5 Implementation of the *ACTB^TK-mPlum* safety switch. a** *NANOG^iCasp9-YFP*;*ACTB^TK-mPlum* hESCs were treated with ganciclovir (or left untreated) for 24 h, and subsequently ganciclovir was withdrawn and hESCs were cultured for 1 week in mTeSR1 to detect any surviving cells and to allow them to regrow. Alkaline phosphatase staining of whole wells of a 12-well plate for each condition (*top*) and cell counts for wild type and *NANOG^iCasp9-YFP*;*ACTB^TK-mPlum* hESCs after each treatment (*bottom*) demonstrated the elimination of hESCs. N.D. = not detected. **b** *NANOG^iCasp9-YFP*;*ACTB^TK-mPlum* hESCs were differentiated into day 6 liver, sclerotome, and neural progenitors and treated with ganciclovir at the indicated doses for the last 24 h of differentiation. Cell survival was analyzed by counting at day 6 of differentiation. **c** $10^6$ *NANOG^iCasp9-YFP*;*ACTB^TK-mPlum* hESCs engineered to express *CAG-AkaLuciferase* were treated with control media or 1 nM AP20187 for 24 h, and then subcutaneously transplanted into the right and left dorsal flanks of NOD-SCID *Il2rg^−/−* mice ($10^6$ cells per flank). After 3 weeks post-transplant, teratomas formed in vivo and ganciclovir was administered daily at 50 mg/kg until week 7 post-transplant. Bioluminescent imaging of mice was conducted weekly for 7 weeks. Total flux (photons/s) was measured for each animal and were averaged (bioluminescent quantification and images for each individual animal from week 0–7 are in Supplementary Fig. 4c). Source data are available in the Source Data file.

We demonstrated that the ACTB^OiCasp9-mPlum kill-switch was functional and effective in vivo, and that it was orthogonal to the NANOG^iCasp9-YFP system. To mimic an adverse event wherein the hPSC-derived cell product had to be destroyed, $10^6$ *ACTB^OiCasp9-mPlum*;*NANOG^iCasp9-YFP* hPSCs were subcutaneously transplanted into the left and right dorsal flanks of NSG mice to form teratomas over the course of 4 weeks (Fig. 6f, Supplementary Fig. 5e and f). 4 weeks post-transplantation, we delivered a single in vivo injection of AP21 (which activates ACTB^OiCasp9-mPlum), which completely eliminated any detectable teratomas within 3 days (Fig. 6f, Supplementary Fig. 5e and f). Even after 4 further weeks, teratomas did not re-emerge, showing that this single AP21 injection fully eliminated teratomas within

the sensitive limit of bioluminescence detection (Fig. 6f; Supplementary Fig. 5e and f). Conversely, in vitro treatment of *ACTB^OiCasp9-mPlum*;*NANOG^iCasp9-YFP* hPSCs with AP20 (which activates NANOG^iCasp9-YFP) killed undifferentiated hPSCs in vitro, and thus prevented teratomas from forming at all (Fig. 6f; Supplementary Fig. 5e and f).

Taken together, the OiCaspase9 cell-ablation system is superior to the aforementioned TK system in two major ways. First, the ACTB^OiCasp9-mPlum safeguard eliminated teratomas more rapidly in vivo (~3 days, with a single AP21 injection; Fig. 6f; Supplementary Fig. 5e and f) than the ACTB^TK-mPlum system (~4 weeks, with daily ganciclovir injections; Fig. 5c; Supplementary Fig. 4c). This can be ascribed to their differing mechanisms-

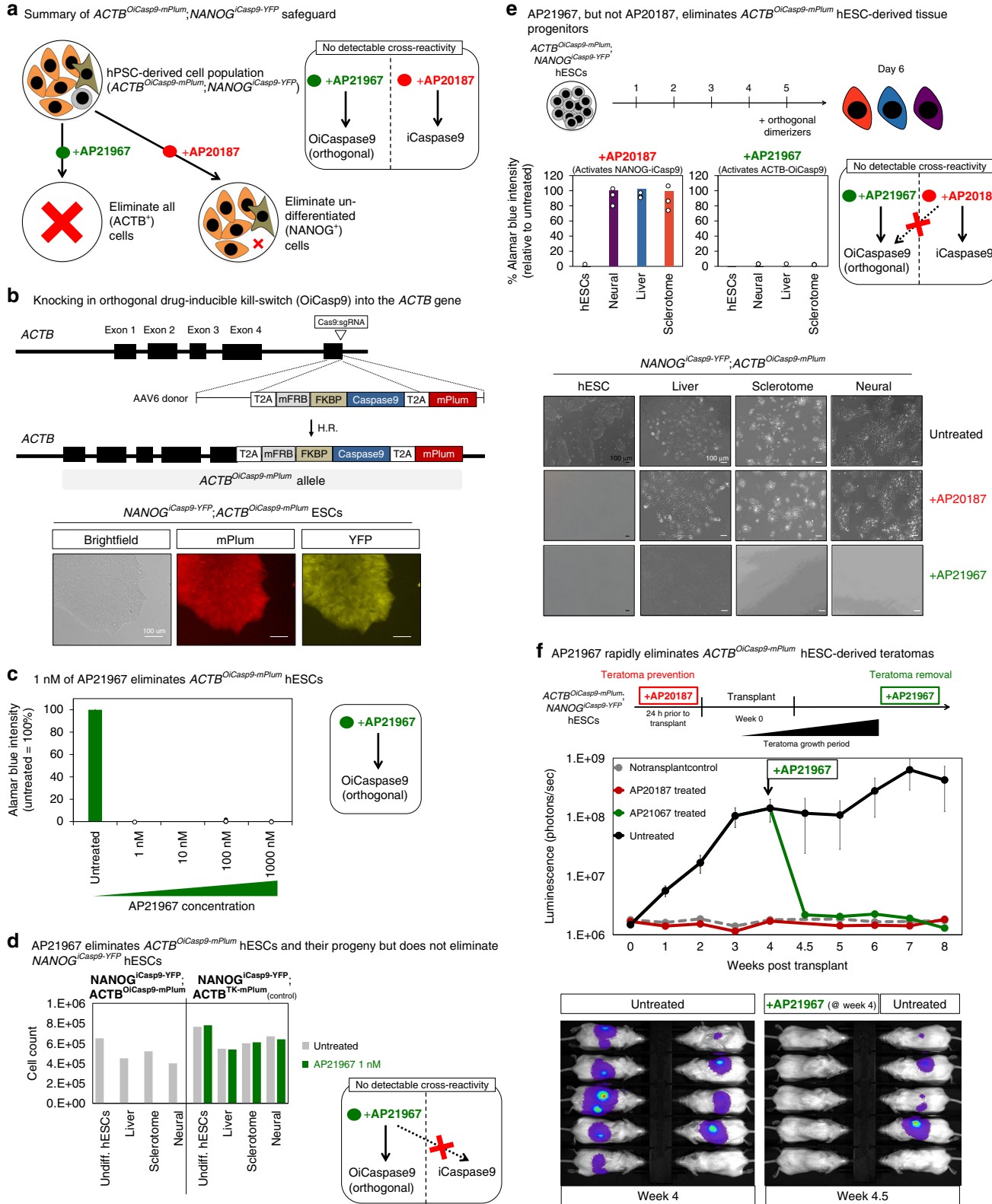

**a** Summary of *ACTB^OiCasp9-mPlum*;*NANOG^iCasp9-YFP* safeguard

**b** Knocking in orthogonal drug-inducible kill-switch (OiCasp9) into the *ACTB* gene

**c** 1 nM of AP21967 eliminates *ACTB^OiCasp9-mPlum* hESCs

**d** AP21967 eliminates *ACTB^OiCasp9-mPlum* hESCs and their progeny but does not eliminate *NANOG^iCasp9-YFP* hESCs

**e** AP21967, but not AP20187, eliminates *ACTB^OiCasp9-mPlum* hESC-derived tissue progenitors

**f** AP21967 rapidly eliminates *ACTB^OiCasp9-mPlum* hESC-derived teratomas

of-action: OiCaspase9 activates the apoptotic pathway to rapidly kill cells, whereas TK inhibits DNA replication, thus principally killing only dividing cells. Second, OiCaspase9 comprises native human proteins and thus should not be immunogenic. This contrasts with the viral protein TK[44]; patients have immunologically rejected TK-expressing cell therapies[49].

## Discussion

Improving the safety of hPSC-derived cell therapies is an important priority in order to make such therapies available to a broad range of patients for diverse indications[3], including those diseases (e.g., non-oncologic diseases) with current therapies that work but are not ideal, in which minimizing risk of a hPSC-derived therapy is

**Fig. 6 Rationale, design, and implementation of the $ACTB^{OiCasp9-mPlum}$ safety switch. a** Intended application of the dual $ACTB^{OiCasp9-mPlum}$; $NANOG^{iCasp9-YFP}$ safeguard. **b** Cas9 RNP/AAV6-based knock-in strategy[38] for $ACTB^{OiCasp9-mPlum}$ targeting, which was performed in the $NANOG^{iCasp9-YFP}$ hESC line. Scale bar = 100 μm. **c** $5 \times 10^5$ $ACTB^{OiCasp9-mPlum}$;$NANOG^{iCasp9-YFP}$ hESCs were treated with control media or 1–1000 nM AP21967 for 24 h. Cell viability was analyzed using alamar blue. Error bars = standard deviation. **d** $ACTB^{OiCasp9-mPlum}$;$NANOG^{iCasp9-YFP}$ and $ACTB^{TK-mPlum}$;$NANOG^{iCasp9-YFP}$ hESCs (negative control), in addition to their respective differentiated derivatives, were treated for 24 h with 1 nM of AP21967. **e** $5 \times 10^5$ $ACTB^{OiCasp9-mPlum}$; $NANOG^{iCasp9-YFP}$ hESCs, and their respective differentiated derivatives, were treated with control media, 1 nM of AP21967 or and 1 nM of AP20187 for 24 h. Surviving cells were analyzed by alamar blue. Scale bar = 100 μm. Error bars = standard deviation. **f** $10^6$ $ACTB^{OiCasp9-mPlum}$;$NANOG^{iCasp9-YFP}$ hESCs engineered to express $CAG$-$AkaLuciferase$ were treated with control media or 1 nM AP20187 for 24 h, and then subcutaneously transplanted into the right and left dorsal flanks of NOD-SCID $Il2rg^{-/-}$ mice ($10^6$ cells per flank). 4 weeks post-transplant, teratomas formed and AP21967 was intraperitoneally administered once at 10 mg/kg. Bioluminescent imaging of mice was conducted weekly for 8 weeks (with the exception of week 4, when imaging was performed again 3 days post-AP21967 administration). Total flux (photons/s) was measured for each animal and was averaged (bioluminescent quantification and images for each individual animal from week 0 to 8 are in Supplementary Fig. 5e and f). Error bars = standard deviation. Source data are available in the Source Data file.

essential. In preclinical models, hPSC-derived cell populations have been reported to form teratomas[8–11], other types of tumor[12–14], or cysts[15] with varying frequencies. Another safety risk is embodied by the tendency of hPSCs to acquire certain genetic mutations upon culture[16–18]; such mutations have been shown to lead to tumor formation by their differentiated progeny in vivo[13]. A less recognized but still important potential application of therapeutic safeguards is for hypoimmunogenic hPSC-based cell products[19,20]. Upon transplantation, if hypoimmunogenic cells become cancerous or infected, they may not be adequately controlled by patients' immune systems. Here we report a general platform to improve the potential safety of hPSC-derived cell therapies, with the aim of mitigating two safety risks that beleaguer this otherwise-promising family of cell therapies.

We engineered hPSCs with dual orthogonal safeguards with different functionalities and that can be activated in response to different contingencies. The first safety switch ($NANOG^{iCasp9}$) provides a method to substantially reduce the risk of teratoma formation *prior* to transplantation. Specifically, it enables the in vitro depletion of teratoma-forming cells from a therapeutic hPSC-derived cell product by >$10^6$-fold using the AP20 drug. This degree of depletion would create a safety buffer for cell products of >1 billion cells or more to be transplanted without the potential toxicity of teratoma formation (given that 10,000 hESCs are sufficient to form a teratoma in mouse models[6]).

By contrast, the second orthogonal safety switch (either $ACTB^{TK}$ or $ACTB^{OiCasp9}$) provides two different ways (treatment with GCV or AP21, respectively) to rapidly eliminate the entire cell product—not just pluripotent cells—if needed. The $ACTB^{OiCasp9}$ system more rapidly eliminates cells in vivo and is likely less immunogenic than the virus-derived $ACTB^{TK}$ system[49]. One might choose to eliminate the cell product because it either had led to adverse events or because it had served its therapeutic purpose and was no longer needed.

Importantly, both of these orthogonal safety switches are integrated into the same hPSC line (the $ACTB^{TK-mPlum}$;$NANO$-$G^{iCasp9-YFP}$ hPSC line and $ACTB^{TK-mPlum}$;$NANOG^{iCasp9-YFP}$ hPSC line), allowing us to use different small molecules to activate distinct safeguards in response to different contingencies. The drugs used to activate our safety switches we describe (ganciclovir and AP20) are safely used in patients[44,50], suggesting the clinical translatability of our proposed safety assurance systems.

Any marker-based strategy to deplete unwanted cells—such as the ones we report here—are only as effective as the specificity of the chosen marker gene. $NANOG$ is a highly specific marker for pluripotent cells in vivo[34–36], which we also show here in vitro. Nonetheless, $NANOG$ is still expressed in rare differentiated lineages (e.g., primordial germ cells[51]). The $NANOG^{iCasp9}$ safeguard, therefore, would not be applicable to such hPSC-derived

cell products. Recently a $SOX2^{iCasp9}$ system was developed to eliminate pluripotent cells[28]. However $SOX2$ is a less-specific marker of pluripotent cells, as it is also expressed in a variety of differentiated lineages including ectodermal (e.g., neural)[52] and anterior endoderm (e.g., liver)[53] progenitors (Fig. 2b). A $TERT^{TK}$ system was also developed to eliminate unwanted, hPSC-derived lineages[24], but this system would only be effective in the specialized scenario that the unwanted cell-type was $TERT^+$ but the desired, therapeutic cell-type was $TERT^-$. A variety of non-genetic means[5] also exist to ablate pluripotent cells, including small molecules (targeting $SCD1$ and $SURVIVIN$)[25–27], cytotoxic antibodies (targeting $PODXL1$)[22], and culture media[54,55] as well as extracellular matrices[56] that preferentially eliminate pluripotent cells. However, $SCD1$, $SURVIVIN$, and $PODXL1$ are expressed in both undifferentiated and differentiated hPSCs (Supplementary Fig. 1a), underscoring the importance of specificity for a cell-ablation system.

The dual orthogonal genome-edited hPSC lines we generated are not suitable for clinical use because they contain foreign fluorescent protein markers and were not manufactured using Good Manufacturing Practices. However such safeguards can be engineered into other hPSC lines with clinically relevant markers (such as truncated versions of NGFR, EGFR, CD19, or CD20) because the Cas9 RNP/AAV6 system we used to genetically engineer these lines is highly efficient and specific across a range of hPSC lines[38]. Moreover, the RNP/AAV6 genome-editing system is so efficient in hPSCs[38] that selectable markers might not even be needed to efficiently identify clones containing biallelic integrations of both of these safeguard systems. While the use of dual safeguards address two important safety concerns for hPSC-derived cell therapies, it would also be possible to engineer cells by genome editing using only one of the systems as well, as they are independent of each other and utilize different genetic loci for their activity.

Finally, our safety systems are precisely knocked into endogenous loci within hPSCs (by contrast to past efforts to randomly insert them using lentiviral transgenes[45,46]), thus reducing the risk of insertional mutagenesis or ectopic silencing of these safety systems. Avoiding transgene silencing should enhance the efficacy of the safeguard system, and avoiding insertional mutagenesis should provide additional safety to the genetically engineered cell product.

## Methods

**hPSC culture**. H9 hPSCs (WiCell)[57] were used throughout this entire study, and were validated to be mycoplasma negative and karyotypically normal. Undifferentiated hPSCs were cultured feeder-free in mTeSR1 media (StemCell Technologies) on cell-culture plates that had been pre-coated with Matrigel (Corning) or Geltrex (Gibco) basement membrane matrices. Each day, mTeSR1 media was changed and hPSC cultures were visually inspected with care to avoid any spontaneous differentiation.

When partially confluent, hPSCs were serially passaged as small clumps by removing mTeSR1, and then adding an EDTA solution (Versene [Gibco]) for 7 min at room temperature to partially dissociate them. Subsequently, EDTA was removed, fresh mTeSR1 was added, and then hPSCs were scraped off of the plate using a cell scraper and then transferred to new Matrigel-coated or Geltrex-coated plates in mTeSR1 media, in accord with WiCell's Feeder Independent culture protocol.

**Genome editing of hPSCs.** To engineer H9 hPSCs carrying the NANOG^iCasp9-YFP, ACTB^TK-mPlum or ACTB^OiCasp9-mPlum systems, we inserted genetic cassettes into the endogenous NANOG or ACTB genes using the Cas9 RNP (ribonucleoprotein)/AAV6 strategy[38]. The main conceit of the Cas9 RNP/AAV6 strategy is to electroporate hPSCs with RNP complexes carrying an engineered, high-specificity HiFi Cas9 variant[58] (Integrated DNA Technologies) complexed with chemically modified sgRNAs[59] (Synthego); simultaneously, AAV6 vectors carrying genetic templates for homologous recombination are also concurrently delivered[38].

To this end, 24 h prior to editing, H9 hPSCs were first treated with 10 μM ROCK inhibitor (Y-27632) to enhance their survival upon future single-cell dissociation. Second, hPSCs at 70–80% confluence were dissociated into single cells by incubating them with Accutase (Life Technologies) for several minutes at 37 °C. Then, ROCK inhibitor-supplemented mTeSR1 media was added to neutralize the dissociating agent. Afterwards, dissociated hPSCs were counted.

Concurrent to cell dissociation, the Cas9 RNP complex was prepared[38]. The RNP complex was formed by combining 5 μg of HiFi Cas9[58] (Integrated DNA Technologies) and 1.75 μg of sgRNA for 10 min at room temperature, which was then diluted with 20 μL of P3 Primary Cell Solution (Lonza).

To electroporate Cas9 RNP complex into hPSCs, 500,000 hPSCs were mixed with the nucleofection solution containing the aforementioned Cas9/sgRNA RNP. hPSCs were then electroporated in a 16-well Nucleocuvette Strip, using the 4D Nucleofector system (Lonza) with the CA137 electroporation code[38]. Following electroporation, cells were plated into one well of a Matrigel-coated 24-well plate containing 500 μL of mTeSR1 media supplemented with 10 μM Y-27632. AAV6 donor vector containing donor constructs for homologous recombination was then directly added to the hPSCs at a 100K multiplicity of infection (MOI). Cells were then incubated with AAV6 donor vector at 37 °C for 24 h. mTeSR1 + 10 μM Y27632 was changed 24 h post-editing; after 48 h, mTeSR1 alone was used (without Y27632).

After Cas9 RNP/AAV6 editing, single hPSCs were expanded as clonal lines for genomic sequencing to confirm successful knock-ins. Altogether, we engineered three safety systems: NANOG^iCasp9-YFP, ACTB^TK-mPlum, and ACTB^OiCasp9-mPlum in H9 hPSCs.

**sgRNA for genome editing.** The NANOG and ACTB synthetic sgRNAs were purchased from Synthego with chemically modified nucleotides at the three terminal positions at both the 5′ and 3′ ends[59]. Modified nucleotides contained 2′-O-methyl 3′-phosphorothioate[59]. sgRNAs were designed to target the ends of the NANOG and ACTB-coding sequences. The genomic sgRNA target sequences, with the PAM sequence in bold, were:

NANOG: 5′-ACTCATCTTCACACGTCTTC**AGG**-3′
ACTB: 5′-CCGCCTAGAAGCATTTGCGG**CGG**-3′

**Construction of donor vectors for homologous recombination.** Genetic cassettes encoding the respective safety switches and flanking homology arms for homologous recombination (NANOG^iCasp9-YFP, ACTB^TK-mPlum, and ACTB^OiCasp9-mPlum) were cloned into the pAAV-MCS plasmid (Agilent Technologies), which contains AAV2 inverted terminal repeat (ITR) sequences. Vectors were designed to replace the stop codon of each respective gene (NANOG or ACTB) and to insert each respective safety switch system immediately downstream of the coding sequence of each gene, in lieu of the stop codon. DNA cloning was performed using the NEBuilder® HiFi DNA Assembly Cloning Kit to create donor vectors for homologous recombination. DNA sequences are provided for the NANOG^iCasp9-YFP construct (Supplementary Data 1), ACTB^TK-mPlum construct (Supplementary Data 2), ACTB^OiCasp9-mPlum construct (Supplementary Data 3) as well as the backbone vector (Supplementary Data 4). Note that while the donor vector backbone for homologous recombination contains AAV2 ITR sequences (Supplementary Data 4), it is pseudotyped with AAV6 capsid proteins (described below) to produce AAV6 particles[38].

**Production of AAV6 donor vectors.** Plasmids were grown in E. coli (NEB® Stable Competent E. Coli (Cat# C3040I) and produced using Invitrogen's Endotoxin-Free Maxi Plasmid Purification Kit (Cat# A33073). Following DNA purification, 50 million 293FT cells (Life Technologies) were plated in 15 cm² dishes. The cells were transfected the next day via 120 μL (1 mg/mL) of PEI (MW 25K) (Polysciences), 6 μg of donor plasmid for homologous recombination (described above), and 22 μg pDGM6 (which carried AAV6 capsid [cap] gene, AAV2 replication [rep] gene, and adenoviral helper genes) (gift from D. Russell)[38]. 72 h post-transfection, cells were harvested and purified using the Takara AAVpro Purification Kit (Cat. 6666) according to the manufacturer's protocol. AAV6 vector titer was determined using digital droplet (ddPCR) to measure vector genome concentration.

**Seeding hPSCs for directed differentiation.** hPSCs were grown to near-confluency at which point they were dissociated into single cells or very small clumps using Accutase (Gibco). Cells were seeded onto Matrigel-coated or Geltrex-coated 12-well plates at a density of ~25,000 cells/cm² in mTeSR1 supplemented with the ROCK inhibitor thiazovivin (1 μM, Tocris; to enable the survival of dissociated hPSCs). The next day after seeding, cells were washed once with DMEM/F12 (to remove all traces of mTeSR1 media) and subsequently, differentiation media was added.

Differentiation media—whose composition is detailed below—was changed every 24 h. Whenever the new differentiation media composition was different from that of the previous day, the cells were briefly washed with DMEM/F12 (to remove any trace of the previous differentiation signals) before adding the new differentiation media.

**hPSC differentiation into liver bud progenitors.** hPSCs were sequentially differentiated towards anteriormost primitive streak, definitive endoderm, and then liver bud progenitors within 6 days with the following media compositions on each day of differentiation[29,30]:

Day 1: CDM2 base media supplemented with 100 ng/mL Activin A + 3 μM CHIR99021 + 20 ng/mL FGF2 + 50 nM PI-103.

Day 2: CDM2 base media supplemented with 100 ng/mL Activin A + 250 nM LDN-193189 + 50 nM PI-103.

Day 3: CDM3 base media supplemented with 20 ng/mL FGF2 + 30 ng/mL BMP4 + 75 nM TTNPB + 1 μM A-83-01.

Day 4–6: CDM3 base media supplemented with 10 ng/mL Activin A + 30 ng/mL BMP4 + 1 μM Forskolin.

The composition of CDM2 basal medium[29,31] is: 50% IMDM + GlutaMAX (Thermo Fisher, 31980-097) + 50% F12 + GlutaMAX (Thermo Fisher, 31765-092) + 1 mg/mL polyvinyl alcohol (Sigma, P8136-250G) + 1% v/v chemically defined lipid concentrate (Thermo Fisher, 11905-031) + 450 μM 1-thioglycerol (Sigma, M6145-100ML) + 0.7 μg/mL recombinant human insulin (Sigma, 11376497001) + 15 μg/mL human transferrin (Sigma, 10652202001) + 1% v/v penicillin/streptomycin (Thermo Fisher, 15070-063). Polyvinyl alcohol was brought into suspension by gentle warming and magnetic stirring, and the media was sterilely filtered (through a 0.22 μm filter) prior to use.

The composition of CDM3 basal medium[30] is: 45% IMDM + GlutaMAX (Thermo Fisher, 31980-097) + 45% F12 + GlutaMAX (Thermo Fisher, 31765-092) + 10% KnockOut serum replacement (Thermo Fisher, 10828028) + 1 mg/mL polyvinyl alcohol (Sigma, P8136-250G) + 1% v/v chemically defined lipid concentrate (Thermo Fisher, 11905-031) + 1% v/v penicillin/streptomycin (Thermo Fisher, 15070-063). Polyvinyl alcohol was brought into suspension by gentle warming and magnetic stirring, and the media was sterilely filtered (through a 0.22 μm filter) prior to use.

**hPSC differentiation into bone progenitors.** hPSCs were sequentially differentiated towards anterior primitive streak, paraxial mesoderm, and sclerotome (bone) progenitors within 6 days with the following media compositions on each day of differentiation[31]:

Day 1: CDM2 base media supplemented with 30 ng/mL Activin A + 4 μM CHIR99021 + 20 ng/mL FGF2 + 100 nM PIK90.

Day 2: CDM2 base media supplemented with 1 μM A83-01 + 250 nM LDN-193189 + 3 μM CHIR99021 + 20 ng/mL FGF2.

Day 3: CDM2 base media supplemented with 1 μM A83-01 + 250 nM LDN-193189 + 1 μM XAV939 + 500 nM PD0325901.

Day 4–6: CDM2 base media supplemented with 1 μM C59 + 5 nM SAG 21K.

**hPSC differentiation into forebrain progenitors.** hPSCs were differentiated into forebrain progenitors with the following media composition for all 6 days of differentiation[32]:

Days 1–6: DMEM/F12 supplemented with 1% N2 (Gibco) + 1% B27 without RA (Gibco) + 1% GlutaMAX (Gibco) + 500 nM LDN-193189 + 3 μM SB-431542 + 1 μM XAV939.

**Teratoma formation.** 10 million NANOG^iCasp9-YFP hPSCs (or various other genetically modified hPSC lines) were seeded in a Geltrex-coated 15-cm dish, in mTeSR1 supplemented with 1 μM thiazovivin (and, when applicable, 1 nM AP20187). 24 h later, cells were then dissociated by treatment with TrypLE Express for 5 min at 37 °C. Dissociated cells in TrypLE Express were diluted 1:10 in DMEM/F12, pelleted and resuspended in 1 mL of a 1:1 mixture of mTeSR1 and Matrigel per original 15-cm dish (~10,000 cells/μL for untreated groups). Tubes were kept on ice until transplant. 6–10-week-old immunodeficient NOD-SCID Il2rg^−/− (NSG) mice—both males and females—were used for all experiments. Mice were anesthetized during transplantation using isoflurane. 100 μL of cell suspension (~1 million cells) was injected subcutaneously into each of the right and left dorsal flanks of the mouse. Teratoma growth was monitored throughout the duration of the experiment via visual inspection and bioluminescent imaging.

**Activation of NANOG^iCasp9-YFP safety system.** The NANOG^iCasp9-YFP safety system was activated by administering AP20187 to uniform populations of a single

cell-type, or alternatively, heterogeneous cell populations containing multiple cell-types.

In the first scenario, uniform populations of $NANOG^{iCasp9-YFP}$ hPSCs or their differentiated progeny were treated with AP20187 (1 nM, or other doses as indicated) for 24 h to deplete pluripotent cells in vitro. AP20187 was added to the appropriate media for each cell-type: mTeSR1 for undifferentiated hPSCs or differentiation media for differentiated cells (composition of differentiation media is described above). Depletion of hPSCs was then quantified by multiple in vivo and in vitro assays, as described below.

In the second scenario, we used AP20187 to kill undifferentiated hPSCs within a heterogeneous cell population. To this end, we simulated a cell-therapy manufacturing error: undifferentiated hPSCs were deliberately spiked into a differentiated cell population. Specifically, $1 \times 10^6$ $NANOG^{iCasp9-YFP}$ hPSCs were dissociated with Accutase and mixed with $1 \times 10^5$ $NANOG^{iCasp9-YFP}$ hPSC-derived day 5 sclerotome cells. This mixed cell population was seeded in sclerotome media (CDM2 base media + 1 μM C59 + 5 nM SAG 21 K [described above]) + 100 ng/ mL FGF2 (to help undifferentiated hPSCs survive) + 10 μM Y-27632 (to help single, dissociated hPSCs adhere and survive), in the presence or absence of 1 nM AP20187 for 1 h. For the remaining 23 h, ROCK inhibitor was removed; that is, the heterogeneous cell populations were cultured in sclerotome media + 100 ng/mL FGF2 in the presence or absence of AP20187.

**Quantifying cell death induced by NANOG^iCasp9-YFP system**. To quantify cell death after AP20187 treatment of undifferentiated or differentiated $NANOG^{iCasp9-YFP}$ hPSCs carrying various safety systems, we used multiple independent assays.

First, we performed a clonal assay for surviving hPSC colonies. To this end, hPSCs were dissociated into single cells with Accutase (Thermo Fisher) and $1 \times 10^6$ cells were plated per well of a six-well plate that was pre-coated with Matrigel. To enhance single-cell survival, hPSCs were plated in mTeSR1 supplemented with ROCK inhibitor 10 μM Y-27632 for 1 h (in the presence or absence of AP20187 at the indicated concentrations). ROCK inhibitor was then withdrawn for the remaining 23 h of culture; that is, hPSCs were cultured in mTeSR1 (in the presence or absence of AP20187). Subsequently, AP20187 was withdrawn altogether and hPSCs were cultured with mTeSR1 for 1 week, to allow any surviving hPSCs to regrow and to form clonal colonies, which were then scored (i.e., 1 surviving colony after AP20187 treatment of $1 \times 10^6$ hPSCs indicated survival of 1 out of $10^6$ cells).

Second, we performed a cell count assay. To this end, hESCs were dissociated with EDTA and $5 \times 10^5$ cells were plated per well of a six-well plate that was pre-coated with Matrigel. Cells were seeded in mTeSR1 + ROCK inhibitor 10 μM Y-27632 (in the presence or absence of AP20187 at the indicated concentrations) for 24 h. Subsequently, cells were dissociated and the number of viable cells were counted using the Bio-Rad TC20™ Automated Cell Counter (trypan blue exclusion).

Third, we performed an Alamar Blue assay for metabolically active cells. To this end, hESCs were cultured in mTeSR1 (in the presence or absence of AP20187 at the indicated concentrations). After 24 h of AP20187 treatment, mTeSR1 media with Alamar Blue (concentration based on manufacturer's protocol) was changed for both untreated and treated samples. A control well containing media + Alamar Blue was used to assess blank wells and to therefore to measure and subtract fluorescence noise.

Fourth, we performed flow cytometric quantification of viable cells. To this end, $NANOG^{iCasp9-YFP}$ hESCs were dissociated into single cells with Accutase and $1 \times 10^6$ cells were plated per well in a six-well plate pre-coated with Matrigel. To enhance single-cell survival, hPSCs were plated in mTeSR1 supplemented with ROCK inhibitor 10 μM Y-27632 for 1 h (in the presence or absence of AP20187 at the indicated concentrations). ROCK inhibitor was then withdrawn for the remaining 23 h of culture; that is, hPSCs were cultured in mTeSR1 (in the presence or absence of AP20187). Subsequently, to quantify the percentage of surviving cells, the cultures were dissociated with TrypLE Express. Cells in TrypLE Express were diluted 1:10 in DMEM/F12 and centrifuged (pelleted) at 500×g for 5 min. Each cell pellet was resuspended in FACS buffer (PBS + 1 mM EDTA [Invitrogen] + 2% v/v FBS [Atlanta Bio] + 1% penicillin/streptomycin [Gibco]) supplemented with DAPI (1:10,000, Biolegend) to discriminate live vs. dead cells. YFP+ (i.e., NANOG+) cells were analyzed (Beckman Coulter CytoFlex Analyzer) to count live cells for both untreated and AP20187-treated groups. In some experiments, YFP+ (i.e., NANOG+) cells were sorted (BD FACS Aria II) and cultured in mTeSR1 to test whether they were actually still living and could form hPSC colonies.

Colony counting assay was performed in seven independent experiments and data shown in Fig. 3a represents the average colony number across all experiments. All other AP20187 treatment experiments for further validation (Alamar Blue proliferation assay, cell counting, and flow cytometry of viable cells) was performed in one independent experiment with three biological replicates assessed per experiment.

**Activation of the ACTB^TK-mPlum safety system**. $ACTB^{TK-mPlum}$;$NANOG^{iCasp9-YFP}$ hPSCs or their differentiated progeny were treated with Ganciclovir (2 μM, or other doses as indicated) for 24 h in vitro. Ganciclovir was added to the appropriate media for each cell-type: mTeSR1 for undifferentiated hPSCs or differentiation media for differentiated cells (composition of differentiation media is described above). To activate the ACTB^TK-mPlum system in vivo, $10^6$ $ACTB^{TK-mPlum}$;

$NANOG^{iCasp9-YFP}$ hPSCs were subcutaneously transplanted into the left and right dorsal flanks of NSG mice (i.e., $10^6$ cells per flank), with the goal of forming teratomas (as described in greater detail above). After 3 weeks of transplantation (during which overt teratomas formed), mice were intraperitoneally treated with ganciclovir (50 mg/kg) daily for 4 additional weeks.

**Quantifying cell death induced by ACTB^TK-mPlum system**. $NANOG^{iCasp9-YFP}$; $ACTB^{TK-mPlum}$ hPSCs ($5 \times 10^5$ cells) were plated and treated with ganciclovir (GCV) at varying concentrations (0.5–2 μM) for 24 h in mTeSR1; subsequently, GCV was withdrawn and hPSCs were cultured in mTeSR1 alone for 6 further days. Three days post-GCV treatment, cell death was observed in hPSCs. At the end of 6 days of culture in mTeSR1 alone, the number of surviving live cells was counted. Each ganciclovir treatment was performed in one independent experiment, with three biological replicates assessed per experiment.

**Activation of the ACTB^OiCasp9-mPlum safety system**. $ACTB^{OiCasp9-mPlum}$; $NANOG^{iCasp9-YFP}$ hPSCs or their differentiated progeny were treated with AP21967 (1 nM, or other doses as indicated) for 24 h in vitro. AP21967 was added to the appropriate media for each cell-type: mTeSR1 for undifferentiated hPSCs or differentiation of cells (composition of differentiation media is described above). To activate the ACTB^OiCasp9-mPlum system in vivo, $10^6$ $ACTB^{OiCasp9-mPlum}$; $NANOG^{iCasp9-YFP}$ hPSCs were subcutaneously transplanted into the left and right dorsal flanks of NSG mice (i.e., $10^6$ cells per flank), with the goal of forming teratomas (as described in greater detail above). After 4 weeks of transplantation (during which overt teratomas formed), mice were intraperitoneally treated with a single dose of AP21967 (10 mg/kg).

**Quantifying cell death induced by ACTB^OiCasp9-mPlum system**. $ACTB^{OiCasp9-mPlum}$; $NANOG^{iCasp9-YFP}$ hPSCs or their differentiated progeny were treated with AP21967 (1 nM or the indicated concentration) for 24 h in the respective culture media (mTeSR1 for hPSCs or respective differentiation media for hPSC-derived differentiated lineages). The number of surviving cells was then quantified by manual cell counting or the alamar blue assay.

**RNA extraction**. To collect RNA for quantitative PCR (qPCR), undifferentiated or differentiated hPSCs were lysed in 350 μL of RLT Plus Buffer and RNA was extracted using the RNeasy Plus Mini Kit (Qiagen) according to the manufacturer's protocol.

**Reverse transcription and qPCR**. Three hundred naograms of total RNA was reverse transcribed into cDNA using the High-Capacity cDNA Reverse Transcription Kit (Applied Biosystems) according to the manufacturer's protocol. qPCR was performed in 384-well format[31] on a QuantStudio 5 qPCR machine (Thermo Fisher). Expression of all genes was first normalized to the levels of the reference gene YWHAZ, and then plotted relative to the levels of YWHAZ (i.e., 1.0 = equivalent to YWHAZ). Forward and reverse primer sequences used to detect the expression of the respective genes are provided in Supplementary Table 1.

All qPCR experiments were performed once. For each qPCR experiment, two biological replicates (two different wells in the same culture plate) were analyzed; qPCR was then performed on two technical replicates for each biological replicate (i.e., 2 biological replicates × 2 technical replicates = 4 replicates altogether). Plotted qPCR data represents the average of the 4 replicates, and standard error is shown.

**Flow cytometry**. Undifferentiated and differentiated hPSCs were dissociated by incubation in TrypLE Express (Gibco) for 5 min at 37 °C. Subsequently, dissociated cells in TrypLE Express were diluted 1:10 in DMEM/F12 and centrifuged (pelleted) at 500×g for 5 min. Each cell pellet was resuspended in FACS buffer (PBS + 1 mM EDTA [Invitrogen] + 2% v/v FBS [Atlanta Bio] + 1% penicillin/streptomycin [Gibco]) supplemented with fluorophore-conjugated primary antibodies. Antibody catalog numbers and staining concentrations are listed in Supplementary Table 2. Antibody staining occurred for 30 min on ice protected from light.

After staining, cells were washed twice with FACS buffer and resuspended in 200 μL FACS buffer with DAPI (1:10,000, Biolegend) for live/dead discrimination. Flow cytometry was performed on a Beckman Coulter CytoFlex analyzer (Stanford Stem Cell Institute FACS Core). For data analysis, cells were gated based on forward and side scatter with height and width used for doublet discrimination. Subsequently, live cells that were negative for DAPI were gated for all marker analyses and calculations of population frequency.

Flow cytometry of undifferentiated hPSCs was performed across three independent experiments, with two biological replicates per experiment (for both Figs. 2d and 4c, respectively). All other flow cytometry experiments were performed once, with two biological replicates per experiment.

**Intracellular flow cytometry**. Undifferentiated hPSCs were dissociated by incubation in TrypLE Express (Gibco) for 5 min at 37 °C. Subsequently, dissociated cells in TrypLE Express were diluted 1:10 in DMEM/F12 and centrifuged (pelleted) at 500×g for 5 min. Cells were subsequently fixed, permeabilized and stained using

the Inside Stain Kit (Miltenyi Biotec #130-090-477) as per the manufacturer's protocol. Antibody catalog numbers and staining concentrations are listed in Supplementary Table 2. Flow cytometry was performed on a Beckman Coulter CytoFlex analyzer (Stanford Stem Cell Institute FACS Core). For data analysis, cells were gated based on forward and side scatter with height and width used for doublet discrimination. Subsequently, cells were gated for all marker analyses and calculations of population frequency.

Each intracellular flow cytometry experiment was performed once, with two biological replicates assessed per experiment.

**Generation of *AkaLuc*-expressing hPSCs.** PiggyBac donor plasmid (pPB-CA-G_AkaLuc_Puro) was constructed by starting from pPB_CAG_rtTAM2_IN (ref. [60]) and then replacing rtTAM2_IN with AkaLuc and Puro, using In-Fusion® HD Cloning Plus (Takara Bio). AkaLuciferase refers a highly sensitive luciferase variant optimized for intravital bioluminescent imaging, which affords the capability to detect single cells in vivo under certain circumstances[43]. PiggyBac transposition was used to deliver the construct into undifferentiated hPSCs for bioluminescent imaging experiments.

**Bioluminescent imaging.** Twenty minutes prior to imaging, mice were injected intraperitoneally with 100 μL of 15 mM AkaLumine HCl (otherwise known as TokeOni [Aobious])[43] dissolved in $H_2O$. Mice were anesthetized using isoflurane and placed in the imaging chamber of either an IVIS Spectrum or SII Lago-X bioluminescent imaging machine. Imaging parameters were kept constant throughout the duration of each experiment with no images reaching saturation (Binning = 4, FStop = 1.2, exposure time = 10 s). Subsequent image analysis was done in Aura with regions of interest (ROIs) drawn for each mouse to calculate total flux (photons/s) in order to quantify teratoma growth over time.

Each bioluminescence imaging experiment was performed once. Taken together, we performed three independent experiments (shown in Figs. 3b, 5c, and 6c):

For the experiment shown in Fig. 3b, four mice were transplanted with hPSCs and four mice were transplanted with AP20187-treated hPSCs.

For the experiment shown in Fig. 5c, 10 mice were transplanted with hPSCs, 10 mice were transplanted with AP20187-treated hPSCs, and 10 mice were transplanted with hPSCs and then treated with Ganciclovir in vivo. Complete data for all mice across all timepoints is shown in Supplementary Fig. 4c.

For the experiment shown in Fig. 6f, five mice were not transplanted, five mice were transplanted with AP20187-treated hPSCs, five mice were transplanted with hPSCs and then treated with AP21967 in vivo and five mice were transplanted with hPSCs and left untreated. Complete data for all mice across all timepoints is shown in Supplementary Fig. 5f.

**Alkaline phosphatase staining.** Alkaline phosphatase staining was done using the Alkaline Phosphatase Staining Kit (Red) (ab242286, Abcam) using the manufacter's protocol. In brief, hPSCs were washed with PBS, fixed, and stained for 20 min using the alkaline phosphatase kit.

**Imaging.** Fluorescent images were taken using the BZ-X710 All-in-One Fluorescence Microscope (Keyence) or the EVOS FL cell imaging system (Thermo Fisher).

**Karyotype analysis.** Karyotype analysis was performed by the Stanford University Cytogenetics Lab. hPSCs growing in Matrigel-coated T25 flasks were dissociated for analysis. Chromosomes were analyzed using the GTW banding method. Twenty metaphase cells were analyzed, all of which were concluded to have a normal karyotype (46, XY). Two karyotyping experiments were performed throughout this study.

**Immunostaining.** hPSCs or their differentiated progeny were fixed in 4% paraformaldehyde for 15 min; permeabilized in 0.2% Triton X-100 in PBS; and then blocked with blocking buffer (0.1% Triton-X and 2% FBS in PBS). For primary staining, anti-NANOG (RRID: AB_10559205), anti-SOX2 (RRID:AB_2195767), and anti-TWIST1 (RRID:AB_883292) antibodies were diluted 200-fold with blocking buffer and stained at 4 °C overnight. Cells were then washed three times and then secondary staining was performed with 1:500 diluted Cy5 Donkey Anti-Rabbit IgG (RRID: AB_2340607) for 1 h. Cells were washed again and DAPI (RRID: AB_2629482) staining was used on the third wash.

For live cells, cells were treated with 5 μg/mL of Hoescht (Invitrogen™ Cat# H3569) for 30 min, washed with PBS, and then imaged using the EVOS FL cell imaging system.

Immunofluorescence experiments were performed once. All catalog numbers for antibodies and other reagents are provided in Supplementary Table 3.

**Genotyping of genetically edited hPSCs.** To confirm successful genetic targeting of the NANOG and ACTB loci, genomic DNA was isolated from NANOG$^{iCasp9-YFP}$; ACTB$^{TK-mPlum}$ hPSCs using QuickExtract DNA Extraction Solution (Epicentre) following the manufacturer's instructions. Then, genomic PCR was performed

using Phusion Green HSII Master Mix (Thermo Fisher) and the primer sequences listed in Supplementary Table 4. For DNA sequencing of the targeted alleles, PCR amplicons were gel-extracted and submitted for Sanger sequencing through MCLab (South San Francisco, CA, USA).

Off-target editing events were predicted for each sgRNA by COSMID46 tool (http://crispr.bme.gatech.edu). Based on these predictions, we identified NANOGP8 as a possible off-target locus and analyzed this possibility using primers detailed in Supplementary Table 4.

Each genomic PCR experiment was performed once, with one biological sample assessed per experiment.

**Regulatory and institutional review.** All animal experiments were conducted in accord with experimental protocols approved by the Stanford Administrative Panel on Laboratory Animal Care (APLAC). All hPSC experiments were conducted in accord with experimental protocols approved by the Stanford Stem Cell Research Oversight (SCRO) committee.

**Statistics.** No statistical methods were used to determine sample size. For animal experiments, no statistical methods were used to determine sample size. In each animal experiment, all animals were analyzed (none were excluded), and data for each individual animal is shown in Figs. 3b, 5c, 6c, Supplementary Fig. 4c, Fig. 6f and Supplementary Fig. 5f. Animals were randomly allocated to experimental groups without pre-selection. When collecting animal data, investigators were not blinded to experimental group assignments.

**Reporting summary.** Further information on research design is available in the Nature Research Reporting Summary linked to this article.

## Data availability

Data supporting the findings of this work are available within the paper and its Supplementary Information files. The datasets generated and analyzed during the current study and any other relevant data are available from the corresponding author upon request. Source data for Figs. 2, 3, 5, and 6 are available in the Source Data file.

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

## Acknowledgements

We thank Volker Wiebking for help with preliminary studies, Timothy Chai and Lay Teng Ang for feedback on the manuscript, and Rayyan Jokhai for assistance with figure preparation. We also thank Porteus and Loh laboratory members for their generous support, as well as the Stanford Institute for Stem Cell Biology & Regenerative Medicine and the Stanford Stem Cell FACS Facility for infrastructure support. R.M.M. is supported by the Stanford Stem Cell Biology and Regenerative Medicine Training Grant (T32GM119995). J.L.F. is supported by National Defense Science and Engineering Graduate (NDSEG) and Stanford Honorary Bio-X Fellowships. M.H.P. is a Chan-Zuckerberg Biohub Investigator and the Laurie Kraus Lacob Faculty Scholar in Pediatric Translational Medicine. Work on this project in the Porteus lab was supported by a philanthropic gift from the Taube/Koret Foundation. K.M.L. is supported by the NIH Director's Early Independence Award (DP5OD024558), Stanford-UC Berkeley Siebel Stem Cell Institute, Stanford Ludwig Center for Cancer Stem Cell Research and Medicine, Stanford Beckman Center and Anonymous Family, and is a Packard Fellow for Science and Engineering, Pew Scholar, Human Frontier Science Program Young Investigator, and The Anthony DiGenova Endowed Faculty Scholar.

## Author contributions

R.M.M., M.K.C., B.J.L., and E.P. genetically engineered hPSC lines carrying safeguard systems. J.L.F. and R.M.M. differentiated hPSCs into specific cell-types and also validated safeguard systems in vivo and in vitro. N.U. and T.N. contributed to preliminary experiments.

## Competing interests

M.H.P. serves on the scientific advisory board for CRISPR Tx and Allogene Tx. Neither company had input into the design, execution, data analysis, or publication of the work presented in this manuscript. N.U. is a current employee of ReGen Med Division, BOCO Silicon Valley. The remaining authors declare no competing interests.
