## [Peer Review File · Nature Communications]

Reviewers' Comments:

Reviewer #1:

Remarks to the Author:

This manuscript by Martin et al describes the genetic targeting of suicide genes into specific loci of iPSc lines, with the objective of making it safer for transplantation approaches. One modification is the knock-in of a caspase9-based kill switch into the NANOG locus, which ensures that pluripotent cells alone express the suicide gene. This results in selective and specific cell death upon in vitro exposure to a pro-drug. A second modification is the introduction of HSV-TK in the actin locus. This is claimed to result in the prevention of cell overgrowth after transplantation. Finally, in another version of their approach, which is presented as an alternative to the latter, a different caspase9, responsive to a different prodrug than the first, is introduced also in the actin locus.

The manuscript has some obvious strengths, including a beautiful presentation of the figures and solid experimental rigor. However, this reviewer encounters several important problems with the rationale for some experiments and the overall novelty of the approach.

Major concerns:

1. Novelty: At the end of the day, the novelty of this manuscript is the use of two different caspase-based suicide switches and their targeted integration into specific loci for endogenous expression. Notable as those technical accomplishments may be, there is no new conceptual advance vs. the state of the art. Others have already reported on the use of CRISPR/Cas9 to integrate suicide genes into human pluripotent stem cell lines (Iwasawa et al, 2019); the integration of more than one suicide gene (Qadir et al, 2019) or the use of inducible caspase9 suicide genes (Yagyu et al, 2015). The selection of the NANOG locus vs. other candidates is well reasoned, but, overall, these authors simply report on a cell line that has two suicide genes. In one of the papers above, one of the suicide genes was used to selectively destroy cells that did not differentiate into the desired lineage. Here, both would simply destroy the entire graft. From this perspective, this does not advance the field significantly.

2. The reviewer is confused about the repeated claim that ganciclovir would just "block the proliferation" of cells expressing HSV-TK. Ganciclovir administration to cells that express HSV-TK kills those cells, period. While it may be argued that only proliferating cells incorporate the nucleotide analog, the net result tends to be complete graft elimination. It can be discussed that this may be due to bystander effect, but the authors' own data show that, when GCV is given to animals that have already well-formed teratomas (i.e., teratomas that would contain not only dividing cells but also terminally different cells), the luminescence of the tumor disappears to "no transplant control" levels (fig. 5c). In other words, in practical terms, both kill switches kill the entire graft. There is no selectivity; only the advantage (already reported by others) that two of such switches are better than one.

Other points (in no specific order):

1. Please explain (or hypothesize) why higher doses of AP20187 are less effective than 1nM (fig. 3a)
2. The experiment in fig. 3b is not, in the view of this reviewer, informative. It is known that the pro-drug kills a vast majority of the cells prior to transplantation. The very few that may survive would probably die anyway if 9 out of 10 surrounding cells are killing bystanders. Therefore, they must be transplanting close to zero live cells after treatment. It is obvious that controls (transplanted with 500,000 undifferentiated cells) will develop teratomas; as it is that, if one transplants zero cells, they won't. They should at least have made an effort to quantify the amount of live cells that they are transplanting in the control group. Or, better still, to compare apples to apples, transplant the exact number of live cells in both groups and then treat one set of animals with AP20 in vivo.
3. Please explain/discuss what the YFP cells that remain after treatment in figure 3d are/could be.
4. Are the suicide genes expressed as a fusion protein along the native protein expressed by the loci they are targeting? If not, please clarify.
5. As stated, it is not correct that any strategies based on pluripotent factors other than NANOG "also deplete the therapeutic product", suggesting that any therapeutic product would also be killed at the

doses that undifferentiated cells die. This is not the case, and the literature abounds in examples (prominently, hTERT)

Reviewer #2:

Remarks to the Author:

Genome edited orthogonal safeguards to improve the safety of human pluripotent stem cell-based therapies

Martin and colleagues showed that hPSCs with "orthogonal safeguards" can efficiently prevent teratoma formation by eliminating remained hPSCs after differentiation and abolishing the abnormally differentiated cells. This approach would be useful to guarantee not only the teratoma free stem cell therapy but also fail-safe against undesired tumorigenic effect of hPSC derived somatic cells.

This study is quite interesting to address two major safety issues of hPSC based cell therapy. The study is well-organized and showed convincing results. A few comments are following.

Major comments

Only very few papers have reported the teratoma formation from hPSC derived cells in the animal models. The reference in the introduction (ref#6) observed one teratoma out of 46 grafts. However, the graft was performed without purification step. This low chance of teratoma formation in animal model may mislead the significance of tumorigenic risk of hPSC based therapy. The author may consider 'host-dependent tumorigenicity' issue and discuss this very briefly in the manuscript (PMID: 12843782). There are more references available to report teratoma/tumor formation of mouse ESCs regardless of purification step in the mouse model (PMID: 23031199, 15557428, 15920163).

There is one reference reporting as low as 245 cells of hESCs developed teratoma in SCID mice (PMID: 19393593).

Other than 'outgrowths' or 'unwanted tissues' to justify 'fail-safe' approach, recent concern of tumorigenicity of somatic cells from hESCs due to 'genetic alteration' of hESCs would be more significant (PMID: 25684226, 22213600).

The concentration of YM155 used in this study toward differentiated cells was very high (2uM). Based on the previous study (PMID:23918355), very low concentration of YM155 (10 – 50nM) was sufficient to eliminate undifferentiated hPSCs. The experiment needs to be performed with 10-50nM range of YM155 (Fig. S1D).

It would be more convincing to provide more systemic approach to select NANOG as 'the most specific to the pluripotent state'. For retinal epithelial cells (RPE), instead of NANOG, of which expression is relatively high in RPE, LIN28A was suggested as a more sensitive marker for pluripotency (PMID: 22615985). Considering the tissue specific gene expression, which may affect the sensitivity to a certain approach for achieving 'selectivity to undifferentiated hPSCs', the limitation of a certain approach depending on the cell type variability needs to be admitted. As NANOG expression is relatively high in the stratified epithelia (PMID: 24979572), it would be challenging to claim that NANOG promoter approach in this manuscript is universal to all type of differentiated cells.

Locus specific KI in hPSCs is technically challenging despite a huge effort to improve the efficiency. How efficient locus specific KI in hPSCs using Cas9 RNP/AAV6 based genome editing in this case?

To exclude the possibility that KI may affect NANOG protein expression, NANOG protein level (with

OCT4, SOX2) needs to be shown with WT control (Fig. S2b).

Complete elimination of any residual undifferentiated hPSCs is essential for this study aim because very few numbers of surviving undifferentiated hPSCs could develop teratoma in vivo. Under the optimal condition (1nM in this case), quantification of live cells determined by FACS (e.g. Annexin V and 7AAD) may be required with proper controls other than measuring intensity of AP, almar blue, so on. As the authors described in the introduction, as low as 10K cells would be risk. For sensitivity to determine residual hPSCs, very sensitive technique to determine the presence of hPSCs by RT-PCR with LIN28A (or NANOG as claimed in this manuscript) is available (PMID: 22615985).

Foreign gene integration into hPSCs sometimes is suppressed. If suicide cassette integrated in NANOG locus is suppressed by unknown reasons and these cells are proliferated, these cells may escape from AP induced elimination process. Thus, it would be important to show the evidence that generation of YFP negative population never occur after several passaging. According to results in figure 3D, a few YFP cells remained survived (1-2%) under AP treatment despite no formation of colony, it would be important to clarify how these cells were survived. This may result from leakage of this system. This should be clarified.

Otherwise, according to recent paper, 'culture adapted hPSCs' due to long term in vitro culture acquire survival advantage to escape the mitochondrial cell death (PMID: 30292852).

Growth completion assay with WT may be necessary as described as previously (PMID: 30292852).

The intact functionality of differentiated cells after elimination of undifferentiated hPSCs would be important issue for practical purpose. Functionality of three different cell types (Liver, Bone and forebrain) should remain unaltered after AP20187 treatment.

According to result in figure 5B, reactivity toward GSV (ganciclovir) appeared to be cell type dependent. It would be because the mode of action of GSV is to inhibition of DNA synthesis, which is critical for actively proliferating cells. Given that, teratoma developed from hPSCs may contain fully differentiated cells that stop proliferating (not all cells in the teratoma would not be actively proliferating). Under such circumstance, complete elimination of teratoma by GSV in figure 5C is less convincing. Similar experiment needs to be performed with fully developed teratoma to see the limited effect of GSV, if any. Otherwise, cell death population in the teratoma by short-course treatment of GSV needs to be shown in the teratoma tissue along with figure 5C.

It is not clear why OiCaspase 9 system as a safeguard system needs to be additionally developed considering highly effective fail-safe role of TK-GSV system in figure 4. What is pro and cons between OiCaspase 9 system and TK-GSV system. If so, it needs to be demonstrated. For example, TK-GSV system may be effective in only actively proliferating unlike OiCaspase 9 system. Fully differentiated neurons would be appropriate model for this experiment.

Results for no-cross reactivity between AP21 and AP20 other than data in figure 6e would be appreciated.

While the usage of AP20187 and AP21967 are to induce dimerization of two proteins (e.g., FKBP or mFRB), it is still crucial to show the cytotoxicity of AP20187 and AP21967 itself on hPSCs. In other words, the cytotoxic effects of both drugs in hPSCs should be validated.

In vivo experiment as similar as that of 5C would be important to claim the Oi-Caspase9 system as an effective fail-safeguard, compared with TK-GSV system.

Overall, this study is well organized and showed convincing results that dual orthogonal system would

be an effective method to guarantee the important safety issue of hPSC based cell therapy. However, as the authors agreed in the text, genetically engineered hPSCs are not acceptable for current clinical application, which would be a major drawback of this approach. This is why a variety of non-genetic approaches such as small molecules, antibodies, fluorescent dye or culture condition has been developed to ablate the tumorigenic cells at the end of differentiation (PMID: 28246701 and 24577362). Notwithstanding, the studies would be still valid to produce hPSCs with dual orthogonal system to ensure the possible tumorigenic risk using gene editing technique in order to minimize the undesirable side effect of gene insertion.

We thank the reviewers and editors for their careful reading and generally favorable reviews. Below we provide a detailed response to each of the concerns raised. By providing additional data (including *in vivo* data) and textual changes, we believe the manuscript is both stronger, more significant and addresses the reviewers' concerns.

Reviewer #1:

This manuscript by Martin et al describes the genetic targeting of suicide genes into specific loci of iPSc lines, with the objective of making it safer for transplantation approaches. One modification is the knock-in of a caspase9-based kill switch into the NANOG locus, which ensures that pluripotent cells alone express the suicide gene. This results in selective and specific cell death upon *in vitro* exposure to a pro-drug. A second modification is the introduction of HSV-TK in the actin locus. This is claimed to result in the prevention of cell overgrowth after transplantation. Finally, in another version of their approach, which is presented as a n alternative to the latter, a different caspase9, responsive to a different prodrug than the first, is introduced also in the actin locus. The manuscript has some obvious strengths, including a beautiful presentation of the figures and solid experimental rigor. However, this reviewer encounters several important problems with the rationale for some experiments and the overall novelty of the approach.

Response: We appreciate the reviewer's positive comments on the figures and "experimental rigor" and "obvious strengths". We also appreciate the problems identified and address these below.

Major concerns:

1. Novelty: At the end of the day, the novelty of this manuscript is the use of two different caspase-based suicide switches and their targeted integration into specific loci for endogenous expression. Notable as those technical accomplishments may be, there is no new conceptual advance vs. the state of the art. Others have already reported on the use of CRISPR/Cas9 to integrate suicide genes into human pluripotent stem cell lines (Iwasawa et al, 2019); the integration of more than one suicide gene (Qadir et al, 2019) or the use of inducible caspase9 suicide genes (Yagyu et al, 2015). The selection of the NANOG locus vs. other candidates is well reasoned, but, overall, these authors simply report on a cell line that has two suicide genes. In one of the papers above, one of the suicide genes was used to selectively destroy cells that did not differentiate into the desired lineage. Here, both would simply destroy the entire graft. From this perspective, this does not advance the field significantly.

Response:

1. We agree with the reviewer that inserting inducible suicide genes into hPSCs via CRISPR/Cas9 is not novel. However, what is new and significant is the **genes that we targeted** (NANOG and ACTB) and the **improved functionalities enabled by our safety systems**, which are **quantitatively more effective than preceding systems**.

Another novelty is our development of **two orthogonal iCaspase9 systems** (iCaspase9 and OiCaspase9), each **activated by a different small molecule**. This permits the activation of two separate safety systems in response to different contingencies. We only presented *in vitro* data on OiCaspase9 in our original submission. However, in our revised paper, we now demonstrate the *in vivo* efficacy of OiCaspase9 (Fig. 6f, Fig. S5e-f) and we confirm that iCaspase9 and OiCaspase9 do not cross-react (Fig. 6d-e, Fig. S5c-d).

2. The reviewer writes "both [safety switches] would simply destroy the entire graft". However, we clarify that **both of our safety switches have different functions** and **do not destroy the entire graft**. They are not redundant:

- **NANOG^{iCaspase9}**: This system **prevents** teratoma formation (it does not kill already-formed teratomas). *In vitro* addition of the activating drug (AP20187) kills undifferentiated hPSCs in culture, **prior to** transplantation. From a pragmatic perspective, we envision that this would be used to kill undifferentiated hPSCs *in vitro* to reduce the risk of teratoma formation upon transplantation.
- **ACTB^{OiCaspase9}** (or **ACTB^{HSV-TK}**): This system **kills the entire cell product (both undifferentiated and differentiated hPSC-derived cell-types)** if an adverse event arises. Addition of the activating drug

(AP21967 or Ganciclovir) kills undifferentiated hPSCs as well as their differentiated progeny, *in vivo* and *in vitro*. From a pragmatic perspective, we envision that this would be used to kill differentiated hPSC-derived cell-types *in vivo* in the event of an adverse event.

We apologize if we did not emphasize the differences between these safety systems sufficiently. We have now edited the Abstract and the Discussion of the paper to make this clearer. We respectfully suggest that our hPSC line that carries **both** of these safety systems is a step forward for the field, as these orthogonal safety systems respectively allow us to 1) prevent teratoma formation ($NANOG^{iCaspase9}$) and 2) eliminate teratomas or other adverse cell populations *in vivo* should they arise ($ACTB^{OiCaspase9}$ or $ACTB^{HSV-TK}$).

2. The reviewer is confused about the repeated claim that ganciclovir would just "block the proliferation" of cells expressing HSV-TK. Ganciclovir administration to cells that express HSV-TK kills those cells, period. While it may be argued that only proliferating cells incorporate the nucleotide analog, the net result tends to be complete graft elimination. It can be discussed that this may be due to bystander effect, but the authors' own data show that, when GCV is given to animals that have already well-formed teratomas (i.e., teratomas that would contain not only dividing cells but also terminally different cells), the luminescence of the tumor disappears to "no transplant control" levels (fig. 5c).

Response: We thank the reviewer for bringing up this point. To address this reviewer's concern, in the revised manuscript we wrote "the TK system [...] principally kills dividing cells (although "bystander" cells may also be indirectly eliminated)" (pg. 7). We appreciate the reviewer's comment as we had many discussions among ourselves about whether we could claim that ganciclovir would kill cells or simply block proliferation. We felt that since we had not demonstrated killing directly (instead only indirectly), we would be cautious in our wording. We appreciate the reviewer's support for using the stronger conclusion and have changed the manuscript accordingly.

In other words, in practical terms, both kill switches kill the entire graft. There is no selectivity; only the advantage (already reported by others) that two of such switches are better than one.

Response: As discussed above, the two different safety systems ($NANOG^{iCaspase9}$ and $ACTB^{OiCaspase9}$) fulfill different functions. The first system kills **undifferentiated hPSCs to prevent teratoma formation** (it cannot kill already-formed teratomas). The second system **kills all hPSC-derived cell-types** (including teratomas) if an adverse event arises.

Regarding "selectivity": $NANOG^{iCaspase9}$ is selective, as it only kills undifferentiated hPSCs, but not their differentiated progeny (**Fig. 3c**). Selectivity is further demonstrated in mixed cultures of hPSCs and differentiated bone progenitors, where the $NANOG^{iCaspase9}$ system specifically kills undifferentiated hPSCs (**Fig. 3d, Fig. S3f**).

We hope this explanation clarifies the power of the $NANOG$ vs. $ACTB$ systems. We agree that the two $ACTB$ systems will kill all cells but the kinetics of killing (especially *in vivo*) is markedly different (see new data in the revised manuscript showing the rapid killing with the $ACTB^{OiCaspase9}$ system).

Other points (in no specific order):

1. Please explain (or hypothesize) why higher doses of AP20187 are less effective than 1nM (fig. 3a).

Response: We thank the reviewer for noticing this. To address this reviewer's concern, in the revised manuscript we wrote "1 nM AP20 was the optimal dose to activate the $NANOG^{iCasp9}$ -YFP system (Fig. 3a), as higher AP20 concentrations downregulated $NANOG$ (Fig. S3b). Indeed, AP20 is structurally related to FK506 (a BMP agonist), and BMP activation is known to downregulate $NANOG$ in hPSCs." (pg. 4).

2. The experiment in fig. 3b is not, in the view of this reviewer, informative. It is known that the pro-drug kills a vast majority of the cells prior to transplantation. The very few that may survive would probably die anyway if 9 out of 10 surrounding cells are killing bystanders. Therefore, they must be transplanting close to zero live cells after treatment. It is obvious that controls (transplanted with 500,000 undifferentiated cells) will develop teratomas; as it is that, if one transplants zero cells, they won't. They should at least have made an effort to quantify the amount of live cells that they are transplanting in the control group.

Response: Our goal was to **prevent** teratoma formation. From a therapeutic point of view, it would be desirable to treat a hPSC-derived cell product with a small molecule **in vitro** to kill off undifferentiated hPSCs prior to transplantation. This would be superior to generating a hPSC-derived cell product, transplanting it *in vivo*, having a teratoma form, and then administering a small molecule as a last measure to stop the teratoma without also eliminating the potentially therapeutic cells that had also been transplanted.

To show that we can **prevent** teratoma formation, in **Fig. 3b** we treated $NANOG^{iCasp9}$ hESCs *in vitro* with AP20187: this eliminated undifferentiated hPSCs and completely prevented them from forming even microscopic teratomas *in vivo* (within the sensitive limit of detection of our AkaLuc-based bioluminescent imaging system).

Or, better still, to compare apples to apples, transplant the exact number of live cells in both groups and then treat one set of animals with AP20 *in vivo*.

Response: We did not test whether AP20187 would kill teratomas *in vivo*, because teratomas largely consist of differentiated cell-types (i.e., not expressing $NANOG$) that constitute the bulk of the tumor. We would not expect, therefore, that *in vivo* AP20187 treatment would ablate hPSC-derived teratomas and we did not make this claim.

However, the reviewer brings up an excellent point: it would nonetheless be important to have a separate system to destroy teratomas *in vivo*. This is exactly what we sought to address with our second kill-switch ($ACTB^{HSV-TK}$ or $ACTB^{OiCasp9}$). In our revision, we include new data on the *in vivo* efficacy of the $ACTB^{OiCasp9}$ kill switch: *in vivo* addition of a small molecule (AP21967) eliminated teratomas in 3 days (**Fig. 6f**; **Fig. S5e,f**).

3. Please explain/discuss what the YFP cells that remain after treatment in figure 3d are/could be.

Response: These residual $NANOG$ - $iCasp9$ -YFP⁺ cells detected in our FACS assay (**Fig. 3di**) are not **functional** hPSCs. This is because, in a functional assay—namely, FACS sorting the YFP⁺ cells and allowing them to grow in mTeSR1 media (in the absence of AP20187)—these YFP⁺ cells did not survive nor did they form colonies within our limit of detection (**Fig. 3dii**). (As a positive control in this experiment, we FACS sorted YFP⁺ undifferentiated hPSCs that were not drug-treated, and they robustly formed colonies [**Fig. 3dii**]). We did not characterize what these cells are and speculate that they might be “zombie” cells that are alive but crippled and non-functional.

4. Are the suicide genes expressed as a fusion protein along the native protein expressed by the loci they are targeting? If not, please clarify.

Response: We apologize for not being clearer. The suicide genes are **not** expressed as fusion proteins.

- In the $NANOG$ -2A- $iCasp9$ -2A-YFP system, the 3 genes are transcriptionally linked, but a T2A self-cleavable peptide separates the three proteins. Therefore a $NANOG$ -2A- $iCasp9$ -2A-YFP mRNA is transcribed, but following translation, the 2A linkers automatically cleave themselves, and 3 separate proteins are produced ($NANOG$, $iCasp9$ and YFP).
- In the $ACTB$ -2A- HSV^{TK} -2A- $mPlum$ system, the same T2A linker strategy is used, such that 3 separate proteins are produced ($ACTB$, HSV^{TK} and $mPlum$).
- In the $ACTB$ -2A- $OiCasp9$ -2A- $mPlum$ system, the same T2A linker strategy is used, such that 3 separate proteins are produced ($ACTB$, $OiCasp9$ and $mPlum$).

The manuscript has been edited to make this explicitly clear (Results section, pgs. 3, 6 and 7).

5. As stated, it is not correct that any strategies based on pluripotent factors other than $NANOG$ “also deplete the therapeutic product”, suggesting that any therapeutic product would also be killed at the doses that undifferentiated cells die. This is not the case, and the literature abounds in examples (prominently, hTERT)

Response: The reviewer raises a good point that any marker-based safety system is only as good as the specificity of the chosen marker gene. This is why we screened the expression of multiple candidate pluripotency marker genes to assess which one(s) were most specific to hPSCs.

In **Fig. S1** and **Fig. 1**, we assessed the expression of multiple candidate pluripotency markers reported in the literature. None of these markers is specific to undifferentiated hPSCs. For instance, *TERT* is reactivated upon late-stage human neural and bone differentiation *in vitro* (**Fig. S1a**). In adult mice, *Tert* is also expressed in a number of adult tissue-restricted stem cells, including intestinal stem cells (Montgomery et al., 2011;

PNAS), liver stem cells (Lin et al., 2018; *Nature*), sperm stem cells (Pech et al., 2015; *Genes & Development*) and hematopoietic stem cells (Morrison et al., 1996; *Immunity*). Therefore eliminating *TERT*⁺ cells would be expected to eliminate undifferentiated hPSCs in addition to certain hPSC-derived tissue stem cells/progenitors (i.e., the therapeutic cell product). Furthermore, our analysis of publicly available microarray data through the NIH database (<https://stemcelldb.nih.gov/>) shows that *TERT* is still expressed during ectodermal as well as mesodermal differentiation of hPSCs (see below):

The reviewer does highlight that if therapeutic cell-type did not express *TERT* but the unwanted, toxic cell-type did express *TERT*, it might be possible to selectively eliminate *TERT*⁺ cells *in vivo* without destroying the therapeutic cell product. For completeness, we have added the following to the discussion:

“A *TERT*^{TK} system was also developed to eliminate unwanted, hPSC-derived lineages²³, but this system would only be effective in the specialized scenario that the unwanted cell-type was *TERT*⁺ but the desired, therapeutic cell-type was *TERT*⁻” (pg. 10).

Reviewer #2:

Genome edited orthogonal safeguards to improve the safety of human pluripotent stem cell-based therapies Martin and colleagues showed that hPSCs with “orthogonal safeguards” can efficiently prevent teratoma formation by eliminating remained hPSCs after differentiation and abolishing the abnormally differentiated cells. This approach would be useful to guarantee not only the teratoma free stem cell therapy but also fail-safe against undesired tumorigenic effect of hPSC derived somatic cells. This study is quite interesting to address two major safety issues of hPSC based cell therapy. The study is well-organized and showed convincing results. A few comments are following.

We thank the reviewer for the positive assessment, and we address the few comments in detail below.

Major comments:

Only very few papers have reported the teratoma formation from hPSC derived cells in the animal models. The reference in the introduction (ref#6) observed one teratoma out of 46 grafts. However, the graft was performed without purification step. This low chance of teratoma formation in animal model may mislead the significance of tumorigenic risk of hPSC based therapy. The author may consider ‘host-dependent tumorigenicity’ issue and discuss this very briefly in the manuscript (PMID: 12843782). There are more references available to report teratoma/tumor formation of mouse ESCs regardless of purification step in the mouse model (PMID: 23031199, 15557428, 15920163). There is one reference reporting as low as 245 cells of hESCs developed teratoma in SCID mice (PMID: 19393593).

Response: We thank the reviewer for these genuinely helpful references, in particular the Hentze et al., 2009 study in *Stem Cell Research* that the reviewer mentioned, and which we now cite in the Introduction of the revised manuscript (pg. 2). Given that our manuscript exclusively regards **human** PSCs, we have focused on the potential risk of human PSC-based therapies, where teratoma formation remains a concern (as discussed in a Lee et al., 2013; *Nature Medicine* review article that we now cite in the revised manuscript).

Other than ‘outgrowths’ or ‘unwanted tissues’ to justify ‘fail-safe’ approach, recent concern of tumorigenicity of somatic cells from hESCs due to ‘genetic alteration’ of hESCs would be more significant (PMID: 25684226, 22213600).

Response: The reviewer brings up the excellent point that hPSCs have been reported to acquire certain genetic abnormalities in culture, which can make them prone to tumor formation. To address the reviewer’s comment, in our revised manuscript we now write: “For example, transplantation of PSC-derived neural populations into animal models generated tumors¹¹⁻¹³ or cysts¹⁴ in some cases. Indeed hPSCs have also been reported to acquire certain genetic abnormalities in culture (e.g., TP53 mutations or BCL2L1 amplifications)¹⁵⁻¹⁷, some of which induce their differentiated progeny to form tumors *in vivo*” (pg. 2).

The concentration of YM155 used in this study toward differentiated cells was very high (2uM). Based on the previous study (PMID:23918355), very low concentration of YM155 (10 – 50nM) was sufficient to eliminate undifferentiated hPSCs. The experiment needs to be performed with 10-50nM range of YM155 (Fig. S1D).

Response: We performed the experiment that the reviewer suggested: 10 nM of YM155 (the SURVIVIN/BIRC5 inhibitor) killed both undifferentiated and differentiated hPSCs (**Fig. S1d**). This is consistent with how *SURVIVIN* mRNA is generally sustained, or even increased, during hPSC differentiation (**Fig. S1a**), explaining why YM155 would also kill differentiated hPSCs.

It would be more convincing to provide more systemic approach to select NANOG as ‘the most specific to the pluripotent state’. For retinal epithelial cells (RPE), instead of NANOG, of which expression is relatively high in RPE, LIN28A was suggested as a more sensitive marker for pluripotency (PMID: 22615985). Considering the tissue specific gene expression, which may affect the sensitivity to a certain approach for achieving ‘selectivity to undifferentiated hPSCs’, the limitation of a certain approach depending on the cell type variability needs to be admitted. As NANOG expression is relatively high in the stratified epithelia (PMID: 24979572), it would be

challenging to claim that NANOG promoter approach in this manuscript is universal to all type of differentiated cells.

Response: The reviewer makes the valuable comment that any marker-based cell depletion strategy is only as good as the specificity of the chosen marker. In the revised Discussion section of our manuscript (Pgs. 9-10), we now write “However, it is important to keep in mind that any marker-based strategies to deplete unwanted cells—such as the ones we report here—are only as effective as the specificity of the chosen marker gene. NANOG is a highly specific marker for pluripotent cells in vivo³³⁻³⁵, which we also show here in vitro. That notwithstanding, NANOG is still expressed in rare differentiated lineages (e.g., primordial germ cells⁵⁰). Therefore the NANOGiCasp9 safeguard would not be applicable to such hPSC-derived cell products.”

We appreciate the comment about the potential for *Nanog* expression in adult mouse stratified epithelia (the Piazzolla et al., 2014; *Nature Communications* paper described by the reviewer), but we note that another study failed to find *NANOG* expression in adult human tissues (Hart et al., 2004; *Dev Dyn*; which we now cite in our revised manuscript).

Locus specific KI in hPSCs is technically challenging despite a huge effort to improve the efficiency. How efficient locus specific KI in hPSCs using Cas9 RNP/AAV6 based genome editing in this case?

Response: Knock-in efficiencies are quantified in **Fig. S5a**.

To exclude the possibility that KI may affect NANOG protein expression, NANOG protein level (with OCT4, SOX2) needs to be shown with WT control (Fig. S2b).

Response: We performed the experiment that the reviewer requested. As shown by intracellular flow cytometry, NANOG protein levels are identical in wild-type and NANOG^{iCasp9-YFP} hPSCs (**Fig. S2c**). In the revised manuscript we now write: “insertion of the iCasp9-YFP cassette downstream of the NANOG gene did not noticeably affect NANOG expression” (pg. 4).

Complete elimination of any residual undifferentiated hPSCs is essential for this study aim because very few numbers of surviving undifferentiated hPSCs could develop teratoma in vivo. Under the optimal condition (1nM in this case), quantification of live cells determined by FACS (e.g. Annexin V and 7AAD) may be required with proper controls other than measuring intensity of AP, almar blue, so on. As the authors described in the introduction, as low as 10K cells would be risk. For sensitivity to determine residual hPSCs, very sensitive technique to determine the presence of hPSCs by RT-PCR with LIN28A (or NANOG as claimed in this manuscript) is available (PMID: 22615985).

Response: We agree with the reviewer about the need for multiple assays to stringently quantify the elimination of undifferentiated hPSCs using the NANOG^{iCasp9} system. While we appreciate the reviewer's suggestion to assess the expression of molecular markers by qPCR (e.g., *LIN28A* or *NANOG*), we believe that the community would ultimately seek a **functional** test of pluripotency. This is because even if qPCR showed a massive depletion of hPSCs, if these residual cells could still form teratomas, ultimately such cells would be therapeutically dangerous. We therefore performed two **functional** tests of pluripotent cells:

- First, we withdrew the AP20187 drug, and continued culture in mTeSR1 medium for 2 weeks, to allow any surviving hESCs to thrive and form colonies. This is a particularly sensitive assay because hESCs divide extremely quickly (once every 15 hours [Zaveri & Dhawan et al., 2018; *Front Cell Dev Biol*]): therefore 2 weeks of culture should allow a single hESC to generate >4 million hESCs (i.e., an extremely obvious colony). In this **functional colony-forming assay**, 1 nM of AP20187 led to a 1.75x10⁶-fold depletion of undifferentiated hPSCs (**Fig. 3a**).
- Second, we withdrew the AP20187 drug and transplanted the cell population into NOD-SCID *Il2rg*^{-/-} mice to test whether they could form teratomas (*N*=3 independent experiments; **Fig. 3b**). This is a particularly sensitive assay because hESCs were engineered to express AkaLuc, and even single AkaLuc⁺ cells can be detected *in vivo* by bioluminescence imaging (Iwano et al., 2018; *Science*). In this functional teratoma-forming assay, in 19/19 mice, **we did not observe teratoma formation**, despite the extremely high resolution afforded by AkaLuc-based detection.

Foreign gene integration into hPSCs sometimes is suppressed. If suicide cassette integrated in NANOG locus is suppressed by unknown reasons and these cells are proliferated, these cells may escape from AP induced elimination process. Thus, it would be important to show the evidence that generation of YFP negative population never occur after several passaging.

Response: We performed the experiment that the reviewer requested. Even after long-term culture (36 passages), the NANOG^{iCasp9-YFP} allele was still ubiquitously expressed by undifferentiated hPSCs (**Fig. S2f**). In the revised manuscript, we now write “hPSCs should not be able to silence this knock-in NANOGiCasp9-YFP system, because if they downregulated endogenous NANOG expression, they would no longer be pluripotent³⁶. Importantly, we inserted the NANOGiCasp9-YFP allele into both NANOG loci to prevent the emergence of “escape” cells (e.g., if a pluripotent cell stochastically used only one allele of NANOG to support its growth⁴⁰)” (pg. 3). Since NANOG is essential for pluripotency, we believe that the bi-allelic knock-in strategy provides a powerful method to mitigate against the development of pluripotent escape variants (through either genetic or epigenetic mechanisms).

According to results in figure 3D, a few YFP cells remained survived (1-2%) under AP treatment despite no formation of colony, it would be important to clarify how these cells were survived.

Response: The reviewer astutely notes that after mixing undifferentiated and differentiated cells and treating them with AP20187, there are “1-2% surviving cells”, as detected by FACS (**Fig. 3d**). Critically, FACS does not show whether the apparently surviving cells are **functional pluripotent cells** or not. Our functional assays (colony formation *in vitro* [**Fig. 3a**] and teratoma formation *in vivo* [**Fig. 3b**]) show that there are at least 10⁶ fold fewer functional hESCs as compared to control conditions. Taken together, our functional colony-forming assay that demonstrates there are no surviving hESCs (within our limit of detection; **Fig. 3dii**) is ultimately a more accurate **functional** assay than the FACS-based assay in quantifying trace numbers of surviving fluorescent positive cells whose identity is unknown.

This may result from leakage of this system. This should be clarified. Otherwise, according to recent paper, ‘culture adapted hPSCs’ due to long term *in vitro* culture acquire survival advantage to escape the mitochondrial cell death (PMID: 30292852).

Response: The reviewer suggests that the NANOG^{iCasp9} system could have “leakage” (i.e., it could be downregulated in pluripotent cells, thus enabling cells to escape the safeguard).

Experimentally speaking, this is not the case. Even after 36 passages, NANOG^{iCasp9} is expressed in all undifferentiated hPSCs (**Fig. S2f**). This therefore shows that the NANOG^{iCasp9} system is not silenced, despite prolonged passaging. We believe that there are 3 reasons why this is the case:

- First, we knocked-in *iCaspase9* into the **endogenous** NANOG gene (in the same transcriptional unit), so therefore any hPSCs that downregulate *iCaspase9* would also no longer express *NANOG* and therefore no longer be pluripotent (e.g., Wang et al., 2012; *Cell Stem Cell*). (This is now mentioned on pg. 3 in the Results.)
- Second, we knocked-in *iCaspase9* cassette into **both alleles** of the *NANOG* gene, so even if a hPSC relied on only one allele of *NANOG* to support its growth (e.g., Miyanari & Torres-Padilla, 2012; *NANOG*), it would still express *iCaspase9*.
- Third, when we performed *NANOG*^{iCasp9} gene targeting, we generated **clonal** hESC lines derived from single hESCs, to exclude the possibility that residual non-targeted hESCs (that do not express *iCasp9*) would persist in the population.

Growth completion assay with WT may be necessary as described as previously (PMID: 30292852).

Response: If we understand correctly, the reviewer is suggesting a growth competition assay to show that in mixed cultures of wild-type vs. kill-switch-containing hESCs, treatment with the small molecule selectively kills the kill-switch-containing hESCs (but not wild-type hESCs). This is now shown in **Fig. S5d**.

The intact functionality of differentiated cells after elimination of undifferentiated hPSCs would be important

issue for practical purpose. Functionality of three different cell types (Liver, Bone and forebrain) should remain unaltered after AP20187 treatment.

Response: We have shown by *qPCR* that the hPSC-derived forebrain, bone and liver progenitors retain expression of their characteristic marker genes despite AP20187 treatment (**Fig. S3e**). In previous work, these marker genes were previously characterized as selective markers of each respective cell-types (Maroof et al., 2013; *Cell Stem Cell*; Loh et al., 2014; *Cell Stem Cell*; Loh et al., 2016; *Cell*; Ang et al., 2018; *Cell Reports*).

These forebrain, bone and liver progenitors are **developmental intermediates** that arise transiently during embryonic development, and **they have no physiological “functionality”** that would otherwise be expected of terminally-differentiated, mature lineages. We thus respectfully believe that testing the “functionality” of these cells is beyond the scope of our present manuscript.

According to result in figure 5B, reactivity toward GSV (ganciclovir) appeared to be cell type dependent. It would be because the mode of action of GSV is to inhibition of DNA synthesis, which is critical for actively proliferating cells. Given that, teratoma developed from hPSCs may contain fully differentiated cells that stop proliferating (not all cells in the teratoma would not be actively proliferating). Under such circumstance, complete elimination of teratoma by GSV in figure 5C is less convincing. Similar experiment needs to be performed with fully developed teratoma to see the limited effect of GSV, if any. Otherwise, cell death population in the teratoma by short-course treatment of GSV needs to be shown in the teratoma tissue along with figure 5C.

Response: The reviewer astutely points out that the HSV^{TK} system does not completely eliminate hPSC-derived differentiated cell populations *in vitro* (**Fig. 5b**). This is because HSV^{TK} produces a nucleotide analog that blocks DNA replication (i.e., it does not actually directly kill cells). To address the reviewer’s concern, we developed the OiCasp9 system to **directly** kill cells by mobilizing the apoptosis pathway. OiCasp9 completely kills hPSC-derived cell populations *in vitro* (**Fig. 6e**, within the limit of detection of our assay), unlike HSV^{TK} (cf. **Fig. 5b**). We note that reviewer 1 was clear in the assessment that ganciclovir would kill cells and not simply cause a proliferation arrest.

It is not clear why OiCaspase 9 system as a safeguard system needs to be additionally developed considering highly effective fail-safe role of TK-GSV system in figure 4. What is pro and cons between OiCaspase 9 system and TK-GSV system. If so, it needs to be demonstrated. For example, TK-GSV system may be effective in only actively proliferating unlike OiCaspase 9 system. Fully differentiated neurons would be appropriate model for this experiment.

Response: We thank the reviewer for appreciating the “highly effective fail-safe role of TK [HSV] system”. As the reviewer requested, we now clearly discuss “pro and cons between OiCaspase 9 system and TK-GSV system” in the revised manuscript:

- “We found that ganciclovir treatment [...] partially eliminated their derivative liver, bone and forebrain progenitors (Fig. 5b). The differing efficacy of ACTB^{TK-mPlum} in undifferentiated and differentiated hPSCs may relate to the differing proliferative rates of these lineages. To eliminate all cells independent of their division rate, we developed a new kill-switch (ACTB^{OiCaspase9}; described in a subsequent section [Fig. 1, Fig. 6]).” (pg. 6)
- “However, the TK system took a prolonged amount of time (~1 month) to eliminate teratomas (Fig. 5c, Fig. S4c) and it principally kills dividing cells (although “bystander” cells may also be indirectly eliminated)⁴³. A new tool to rapidly kill the entire hPSC-derived cell product (not just dividing cells) would thus be advantageous, and is described below.” (pg. 7)
- “Taken together, the OiCaspase9 cell-ablation system is superior to the aforementioned TK system in two major ways. First, the ACTB^{OiCasp9-mPlum} safeguard eliminated teratomas more rapidly *in vivo* (~3 days, with a single AP21 injection; Fig. 6f; Fig. S5e-f) than the ACTB^{TK-mPlum} system (~4 weeks, with daily ganciclovir injections; Fig. 5c; Fig. S4c). This can be ascribed to their differing mechanisms-of-action: OiCaspase9 activates the apoptotic pathway to rapidly kill cells, whereas TK inhibits DNA replication, thus principally killing only dividing cells. Second, OiCaspase9 comprises native human proteins and thus should not be immunogenic. This contrasts with the viral protein TK; patients have immunologically rejected TK-expressing cell therapies⁴⁸.” (pg. 8)

We believe the above addresses the reviewer's concerns and explains the key differences between the two systems (OiCaspase9 and TK).

Results for no-cross reactivity between AP21 and AP20 other than data in figure 6e would be appreciated.

Response: We now present 4 experiments to show that AP20 and AP21 **do not** cross-react. The conclusion of each such experiment is now summarized in cartoon format next to each respective figure:

- **Fig. 6d:** AP21967 does not activate iCaspase9
- **Fig. 6e:** AP20187 does not activate OiCaspase9
- **Fig. S4c:** AP21967 does not activate iCaspase9
- **Fig. S4d:** AP21967 does not activate iCaspase9

While the usage of AP20187 and AP21967 are to induce dimerization of two proteins (e.g., FKBP or mFRB), it is still crucial to show the cytotoxicity of AP20187 and AP21967 itself on hPSCs. In other words, the cytotoxic effects of both drugs in hPSCs should be validated.

Response: We have shown that all 3 drugs (AP20187, AP21967 and Ganciclovir) are not toxic to hPSCs that do not bear the safety system (at the doses used throughout our paper):

- **AP20187:** 1-100 nM AP20187 is not toxic to wild-type hPSCs (**Fig. S3a**).
- **Ganciclovir:** 0.5-2 μ M Ganciclovir is not toxic to wild-type hPSCs (**Fig. 5a**).
- **AP21967:** 1 nM AP21967 is not toxic to hPSCs that lack *OiCasp9* system (**Fig. 6d, Fig. S5c**).

In vivo experiment as similar as that of 5C would be important to claim the Oi-Caspase9 system as an effective fail-safeguard, compared with TK-GSV system.

Response: We have completed the experiment that the reviewer suggested. Notably, we find that the *ACTB^{OiCasp9}* system is even more rapid in eliminating teratomas: with a **single in vivo** treatment of AP21967, **3 days later**, teratomas are completely eliminated as assessed by bioluminescent imaging (i.e., indistinguishable from background signal) (**Fig. 6f, Fig. S5e-f**).

This is superior to our alternative system (*ACTB^{TK}*), which we showed requires **4 weeks of daily in vivo** drug treatment (Ganciclovir) to eliminate teratomas in the same assay (**Fig. 5c**).

Our finding that OiCaspase9 kills teratomas more swiftly than HSV^{TK} is consistent with the molecular mechanisms through which these two systems operate. In our revised manuscript, we now write: "This can be ascribed to their differing mechanisms-of-action: OiCaspase9 activates the apoptotic pathway to rapidly kill cells, whereas TK inhibits DNA replication, thus principally killing only dividing cells" (pg. 8).

Overall, this study is well organized and showed convincing results that dual orthogonal system would be an effective method to guarantee the important safety issue of hPSC based cell therapy. However, as the authors agreed in the text, genetically engineered hPSCs are not acceptable for current clinical application, which would be a major drawback of this approach. This is why a variety of non-genetic approaches such as small molecules, antibodies, fluorescent dye or culture condition has been developed to ablate the tumorigenic cells at the end of differentiation (PMID: 28246701 and 24577362). Notwithstanding, the studies would be still valid to produce hPSCs with dual orthogonal system to ensure the possible tumorigenic risk using gene editing technique in order to minimize the undesirable side effect of gene insertion.

Response: We thank the reviewer for the suggestion. In the expanded Discussion section (pg. 10) we discuss other strategies to deplete undifferentiated hPSCs, including small molecules, antibodies and culture methods.

Reviewers' Comments:

Reviewer #1:

Remarks to the Author:

I think the authors have sufficiently addressed the reviewer's concerns.

Reviewer #2:

Remarks to the Author:

The manuscript was markedly improved and the authors addressed most of the comments from the reviewer. The efficacy of these approaches as either fail-safe or teratoma inhibition was well presented. However, resolving a few minor issues would be necessary.

The authors claimed that there was no clear difference in cell cytotoxicity between hPSCs and differentiated cells even in the low concentration of YM155 (10 nM). However, the viability assay performed in this study (Fig. S1B) was limited to Alamar Blue assay, which we could not discriminate whether the Alamar blue positive populations are actual dead cells or growth arrested cells. Thus, we recommend other cell viability assay (e.g. Trypan Blue or Annexin-V assay) to clarify the different sensitivity of both hPSCs and differentiated cells under low concentrations of YM155 (10-50 nM). As described the first review, intact functionality instead of marker expressions would be still important despite the response from the authors, because the strength of this work is to aim the efficient technique to guarantee the safety of future clinical application. In this point, differentiated cells but not undifferentiated cells should still remain intact in vivo after induction of hPSCs specific cell death, while no teratoma was formed when differentiated and undifferentiated cells from hPSCs are transplanted.

We thank the reviewers and editors for their careful reading and generally positive reviews. Our responses to their remaining comments are below.

Reviewer #1:

I think the authors have sufficiently addressed the reviewer's concerns.

Response: Thank you.

Reviewer #2:

The manuscript was markedly improved and the authors addressed most of the comments from the reviewer. The efficacy of these approaches as either fail-safe or teratoma inhibition was well presented. However, resolving a few minor issues would be necessary. The authors claimed that there was no clear difference in cell cytotoxicity between hPSCs and differentiated cells even in the low concentration of YM155 (10 nM). However, the viability assay performed in this study (Fig. S1B) was limited to Alamar Blue assay, which we could not discriminate whether the Alamar blue positive populations are actual dead cells or growth arrested cells. Thus, we recommend other cell viability assay (e.g. Trypan Blue or Annexin-V assay) to clarify the different sensitivity of both hPSCs and differentiated cells under low concentrations of YM155 (10-50 nM).

Response: If we understand correctly, the Alamar Blue assay detects metabolically active cells, as it involves the use of a redox dye (resazurin) that changes color after reduction within actively respiring cells. To address the reviewer's comment, in our revised manuscript we write that "we found that the small-molecule SURVIVIN inhibitor YM155 (which was previously reported to kill undifferentiated hPSCs^{26,27}) **was inhibitory to** both undifferentiated and differentiated hPSCs" (pg. 3), because technically we did not show that YM155 actually killed the cells (as the reviewer points out).

As described the first review, intact functionality instead of marker expressions would be still important despite the response from the authors, because the strength of this work is to aim the efficient technique to guarantee the safety of future clinical application. In this point, differentiated cells but not undifferentiated cells should still remain intact in vivo after induction of hPSCs specific cell death, while no teratoma was formed when differentiated and undifferentiated cells from hPSCs are transplanted.

Response: We agree with the reviewer's comment that in the future, it will be important to demonstrate that hPSC-derived differentiated cell-types remain functional despite activation of the NANOG^{iCasp9} safeguard (which should selectively eliminate undifferentiated hPSCs). However, we respectfully believe that these experiments are beyond the scope of the present report. In revising the paper, we instead devoted the majority of our efforts in developing and characterizing the new orthogonal iCaspase9 safeguard and implementing it in the form of the ACTB^{OiCasp9} system.